# Leveling Up the Controversial Role of Neutrophils in Cancer: When the Complexity Becomes Entangled

**DOI:** 10.3390/cells10092486

**Published:** 2021-09-20

**Authors:** Ronit Vogt Sionov

**Affiliations:** Hadassah Medical School, The Hebrew University of Jerusalem, Ein Kerem Campus, P.O.B. 12272, Jerusalem 9112102, Israel; Ronit.sionov@mail.huji.ac.il

**Keywords:** cancer, Cathepsin G, chemokines, chronic inflammation, metastasis, NETs, neutrophils, RAGE, reactive oxygen species, TRAIL

## Abstract

Neutrophils are the most abundant immune cell in the circulation of human and act as gatekeepers to discard foreign elements that have entered the body. They are essential in initiating immune responses for eliminating invaders, such as microorganisms and alien particles, as well as to act as immune surveyors of cancer cells, especially during the initial stages of carcinogenesis and for eliminating single metastatic cells in the circulation and in the premetastatic organs. Since neutrophils can secrete a whole range of factors stored in their many granules as well as produce reactive oxygen and nitrogen species upon stimulation, neutrophils may directly or indirectly affect carcinogenesis in both the positive and negative directions. An intricate crosstalk between tumor cells, neutrophils, other immune cells and stromal cells in the microenvironment modulates neutrophil function resulting in both anti- and pro-tumor activities. Both the anti-tumor and pro-tumor activities require chemoattraction towards the tumor cells, neutrophil activation and ROS production. Divergence is seen in other neutrophil properties, including differential secretory repertoire and membrane receptor display. Many of the direct effects of neutrophils on tumor growth and metastases are dependent on tight neutrophil–tumor cell interactions. Among them, the neutrophil Mac-1 interaction with tumor ICAM-1 and the neutrophil L-selectin interaction with tumor-cell sialomucins were found to be involved in the neutrophil-mediated capturing of circulating tumor cells resulting in increased metastatic seeding. On the other hand, the anti-tumor function of neutrophils was found to rely on the interaction between tumor-surface-expressed receptor for advanced glycation end products (RAGE) and Cathepsin G expressed on the neutrophil surface. Intriguingly, these two molecules are also involved in the promotion of tumor growth and metastases. RAGE is upregulated during early inflammation-induced carcinogenesis and was found to be important for sustaining tumor growth and homing at metastatic sites. Cathepsin G was found to be essential for neutrophil-supported lung colonization of cancer cells. These data level up the complexity of the dual role of neutrophils in cancer.

## 1. Introduction

### 1.1. Neutrophil Differentiation in the Bone Marrow

Neutrophils are short-lived white blood cells of the innate immune system that are continuously replenished by newly differentiated cells in the bone-marrow. The neutrophils are differentiated in the bone-marrow from a common myeloid progenitor (CMP) through a common granulocyte-monocyte progenitor (GMP). This is followed by several stages: myeloblasts, promyelocytes, myelocytes, metamyelocytes (also called banded neutrophils or immature neutrophils) and mature neutrophils [1,2,3,4]. During this process, the various granules are formed that store different components, preventing them from interacting with each other within the cell [2,5]. 

The primary azurophilic granules are the first granules to be formed, which occurs in the promyelocyte stage. Then, the secondary specific granules are formed in the myelocyte stage, while the tertiary gelatinase granules are formed in the metamyelocyte stage. The last to be formed is the secretory vesicles, which takes place in the mature segmented neutrophils [2,3,6]. The secretory vesicles and the three different forms of granules are important for enabling rapid changes in neutrophil function upon stimuli. Upon exocytosis, the content is discharged to the surroundings while their membrane components become expressed on the neutrophil cell surface, thereby, enabling the neutrophils to respond promptly to additional stimuli in their surroundings. 

The sequential release occurs in the opposite direction of their formation, meaning that the secretory vesicles are the first to be released, then the tertiary granules, the secondary granules and, at last, the primary granules [3]. The differential release of the various granules of the neutrophils may contribute to the neutrophil heterogeneity, versatility and plasticity observed under various pathophysiological conditions [7,8]. This may also explain why the premature neutrophils released to the circulation exhibit different properties than the mature neutrophils, and why the neutrophils, during their short lifespan, exhibit different traits depending on the stimuli perceived and their stage of differentiation. 

The short half-life of the circulating neutrophils together with the continuous replenishment of neutrophils from the bone-marrow, leads to constant dynamic changes in the neutrophil population. In addition, exposure to different combinations of cytokines and chemokines modulates not only the activation state of the mature neutrophils but also normal and emergency granulopoiesis [9,10].

### 1.2. The Neutrophil Life Cycle

Under healthy conditions, most of the immature neutrophils are retained in the bone-marrow through CXCR4-CXCL12 ligation [11,12]. However, under inflammatory conditions and in cancer, there is an increase in neutrophil mobilization resulting in the release of immature neutrophils into circulation. G-CSF mobilizes neutrophils from the bone-marrow to the circulation by Cathepsin G (CathG)- and Neutrophil elastase (NE)-mediated cleavage of CXCR4 and its ligand CXCL12 [13]. The release of the neutrophils to the circulation is accompanied by an upregulation of CXCR2 [11], which responds to the chemokine CXCL2. CXCL2 is important for neutrophil recruitment both in inflammation and in cancer [14]. 

Freshly released neutrophils express high levels of L-Selectin (CD62L) that is progressively reduced during their lifetime in circulation accompanied by the upregulation of CXCR4 [15,16]. The circulating neutrophils also show heterogeneity in CD11b/CD18 (Mac-1) expression [17], which is important for interaction with endothelial cells [18,19], erythrocytes [17], platelets [17] and T cells [20]. Intact Src kinase function was found to be important for the full activation of the β2 integrins [21] and erythrocyte-neutrophil interaction [17]. The senescent CXCR4-expressing neutrophils respond to the chemokine CXCL12, resulting in their egress from the circulation into the bone marrow where they are eliminated by macrophages [12] with feedback inhibition of neutrophil production through the IL-17/G-CSF axis [22]. 

The increase in CXCR4 expression during neutrophil senescence is inhibited by IFNγ, IFNα, IFNβ, GM-CSF and G-CSF [14,23], resulting in an extended neutrophil longevity and an increase in the number of senescent neutrophils that have acquired different functions than the newly released mature neutrophils [15,24,25]. The most prominent characteristic of senescent neutrophils is their ability to form neutrophil extracellular traps (NETs) [24], which is considered a neutrophil suicide mechanism to capture and kill microorganisms [26]. As will be discussed in Section 4.5, NETs are also involved in capturing circulating tumor cells and promote their growth and metastatic seeding in a feed-forward viscous loop.

### 1.3. General Neutrophil Functions

In general, neutrophils are considered to be immunosurveillance cells whose function is to eliminate any foreign bodies that have penetrated the circulation and the tissues [7,27]. Through their whole battery of membrane receptors and secretory molecules they communicate with other immune cells to elicit co-operating immune responses for combating the invaders. In addition, they interact with both endothelial and epithelial cells and the extracellular matrix that enable their migration into and through the tissues [28,29,30]. 

In addition to these functions, neutrophils play a central role in wound healing [31,32] and in resolving inflammation [33]. Cancer is usually considered as “wounds that do not heal”, and the tumor microenvironment shares several traits with wound healing processes [34,35]. Moreover, tumors are characterized by a chronic inflammatory microenvironment and the immunosuppressive nature of the surroundings is responsible for tumor angiogenesis, progression and invasion [36].

Neutrophils have been recognized to play an important role in cancer. However, different studies have drawn opposite conclusions of the function of neutrophils in tumor progression and metastasis, and many efforts have been made to understand this controversy [37,38,39,40,41,42,43,44,45,46,47,48,49]. This review will discuss different aspects that can catch light on the growing complexity of neutrophils in cancer. First, the evidence for neutrophil involvement in cancer will be discussed followed by neutrophil diversity and plasticity. Next, the mechanisms regulating the pro- and anti-cancer phenotypes and their modes of action will be described. Last, the mechanisms of neutrophil recognition of cancer cell will be highlighted with a specific emphasize on the newly recognized interaction between neutrophil-surface-expressed Cathepsin G with tumor cell expressed RAGE.

## 2. Evidence for Neutrophil Involvement in Cancer

### 2.1. Overview of Tumor Models Showing Pro- versus Anti-Tumor Neutrophil Functions

Some studies claim a pro-tumor function where the neutrophils promote tumor growth and metastasis formation, while, on the contrary, others have attributed an anti-tumor function where the neutrophils prevent tumor progression and metastasis. Several of the conflicting reports seem to lie in the different animal cancer models used, where elimination of neutrophils in certain cancer models leads to reduced metastatic seeding indicative for a pro-metastatic role [50,51,52,53,54,55,56,57,58,59], while similar elimination of neutrophils in other cancer models leads to the opposite, namely an increase in metastases, suggesting for an anti-metastatic activity of neutrophils [60,61,62] (Table 1 and Table 2). 

The conclusions of these studies are based on the use of antibodies that eliminate neutrophils; however, it should be kept in mind that the neutrophils are continuously replenished from the bone-marrow, and the newly released neutrophils, which have resisted anti-Ly6-mediated depletion, may still be functional [63]. There is no possibility to completely remove all neutrophils because this can lead to life-threatening infections. Nevertheless, reducing the neutrophil number in tumor-bearing mice was sufficient to alter the ability of the cancer cells to metastasize, whether it is an increase or a decrease in the metastatic capability. When looking at neutrophils as a discrete cell type, the opposite effects of neutrophil depletion on tumor growth and metastases can, in part, be explained by differential activation of neutrophils by tumor cells and other cells in the tumor milieu, resulting in distinct ratios of pro- versus anti-tumor neutrophils.

There is no simple explanation for these contradictory effects, which are mediated by a complex crosstalk between neutrophils, other immune cells, tumor cells and stromal cells in the tumor microenvironment [64]. As will be discussed in Section 3, the neutrophils in cancer constitute a heterogeneous population of both anti- and pro-tumor neutrophils as well as granulocyte myeloid-derived suppressor cells (G-MDSCs) [7,48,65,66,67]. The ratio and locations of the different neutrophil subpopulations might dictate the net effect of neutrophils on tumor progression and metastasis. The neutrophils in the metastatic site might have different characteristics than those in the circulation and those at the primary tumor site [65,68,69,70]. In some tumors, neutrophils do not affect the primary tumor growth [51,60,71], while in others they do [72,73,74]. For instance, neutrophils isolated from the primary tumor of 4T1 breast carcinoma barely exhibited anti-tumor activities, while those isolated from the lungs of the same animals showed anti-tumor activities to a similar extent as the circulating neutrophils [68]. 

It has been suggested that high levels of TGFβ in the primary tumor prevent the anti-tumor function of neutrophils and promote the appearance of an immunosuppressive neutrophil population [68,75]. The TGFβ level is anticipated to be much lower in the pre-metastatic lung, which thus enables the actions of anti-tumor neutrophils [68]. However, in the MMTV-polyoma middle T antigen (PyMT) mammary tumor mouse model, neutrophil recruitment to the pre-metastatic lung could specifically support metastatic initiation through neutrophil-derived leukotrienes that promotes the growth of a subpopulation of cancer cells [76]. Again, we see that the cancer regulating activities of neutrophils are complex, full of dualities, which will be further discussed in this review.

#### 2.1.1. The Anti-Tumor Activities of Neutrophils Can Be Masked by the Immunosuppressive Activities

Tumor rejection is achieved by a combined effect of direct anti-tumor activity of neutrophils, neutrophil-induced anti-tumor T cell responses and anti-tumor NK cell activities [83]. Neutrophils might also modulate the anti-tumor function of macrophages [84]. However, the simultaneous presence of G-MDSCs that tune down the activities of both cytotoxic T and NK cells might overshadow the anti-tumor neutrophil function in various experimental settings [85,86]. Recently, Li et al. [87] observed that neutrophils have an inhibitory effect on metastatic colonization of breast cancer cells in NK-deficient mice, while facilitating metastatic colonization in NK cell competent mice. They argued that, in both mice, the neutrophils showed anti-tumor activities. However, since the neutrophils suppress the tumoricidal activity of NK cells, the elimination of neutrophils led to recovery of the NK cells, which reduced the metastatic seeding. In this study, the input of NK cells was larger than that of the anti-tumor neutrophils. Thus, the net in vivo effect of neutrophil depletion on cancer metastasis is affected by the anti-tumor activity of other immune cells.

#### 2.1.2. Tumors Secreting G-CSF/GM-CSF Together with Chemokines Induce a Predominant Anti-Tumor Phenotype

Tumor models that showed predominant anti-tumor neutrophil function are characterized by relatively high tumor cell production of CXCL2 and other chemokines together with G-CSF/GM-CSF, resulting in the preferential accumulation of activated neutrophils on the expense of other immune cells [60,61]. Under these conditions, the neutrophils are the major players, and thus neutrophil depletion results in increased tumor growth [60,61]. 

The importance of chemokines in promoting the anti-tumor phenotype is supported by the finding that PyMT-CXCR2^−/−^ neutrophils exhibit reduced tumor killing activity with concomitant increased pro-tumor activities compared to PyMT-CXCR2^+/+^ neutrophils [69]. The picture becomes even more complex when we take into account that the same neutrophil might change its activities during its short lifespan in circulation [67]. The anti-tumor activity is especially attributed to the young mature normal high-density neutrophils (HDN), which can convert into senescent low-density neutrophils (sLDN) that exhibit pro-tumor activity [67]. Thus, factors increasing the longevity of neutrophils might indirectly increase the pro-tumor function.

#### 2.1.3. Neutrophils May Contribute to Immune Exclusion

In a Kras^G12D^-driven mouse model of lung cancer, the tumor was found to be mainly infiltrated by neutrophils, while most of the other immune cells resided outside the tumor mass indicating a state of immune exclusion [81]. Elimination of neutrophils led to enrichment of FasL^low^, MECA-79^high^ endothelial cells in the tumor, which permitted cytotoxic CD8^+^ T cell infiltration with concomitant reduction in T regulatory cells [81]. The neutrophils, through modulation of endothelial cells, caused intra-tumoral hypoxia, and the resulting stabilization of HIF-1α induced Snail expression in the tumor cells [81]. Snail, in turn, led to CXCL5 secretion by the tumors with a concomitant increase in CXCL2 expression in the neutrophils [81]. This crosstalk made the tumor more aggressive [81].

In a Kras^LSL-G12D/+^; Trp53^LSL-R172H/+^; Pdx1-Cre mouse model of pancreatic adenocarcinoma, systemic depletion of GR1^+^ myeloid cells, including neutrophils, increased the infiltration of effector T cells involved in the inhibition of tumor growth [92,93]. CXCR2 inhibition prevented neutrophil accumulation in the pancreatic tumors and led to a T cell-dependent suppression of tumor growth [94]. Mehmeti-Ajradini et al. [95] observed that G-MDSCs isolated from metastatic breast cancer cell patients reduced endothelial expression of CX3CL1 (fractalkine) and prevented infiltration of myeloid immune cells into the tumor. 

The endothelial-derived CX3CL1 has been shown to be responsible for tuning immunologically cold tumor into hot tumor [96]. Its expression is downregulated by the pro-angiogenic growth factors vascular endothelial growth factor (VEGF) and basic fibroblast growth factor (bFGF) [97], which are among others produced by pro-tumor neutrophils or released from the extracellular matrix (ECM) by neutrophil-derived enzymes (see Section 4.4).

### 2.2. Association of High Neutrophil-to-Lymphocyte Ratio (NLR) and Intra-Tumoral Neutrophil Infiltration with Cancer Progression

Several studies have focused on the correlation between neutrophil blood count, neutrophil-to-lymphocyte ratio (NLR) and/or intratumor neutrophil infiltration with the overall survival, remission and/or disease recurrence in cancer patients. The impact of these parameters on tumor progression depends on the cancer type and has been extensively reviewed elsewhere [39,41,42,98]. A high neutrophil-to-lymphocyte ratio (NLR ≥ 4) has often been associated with a poorer overall survival (e.g., ovarian cancer, pancreatic ductal adenocarcinoma, breast cancer, colorectal carcinoma, esophageal cancer, glioblastoma and head and neck cancer) [99,100,101,102,103,104,105,106,107,108,109,110]. 

A prognostic impact of NLR may be due to an association of high NLR with inflammation [98]. A high NLR might also be indicative for a more advanced disease [111,112], as neutrophil count often increases upon disease progression and, in such, might be a reason for a shorter overall survival [41]. A high NLR may also be due to a distorted preferential differentiation of hematopoietic stem cells to the granulocyte lineage and the repression of T cell proliferation and cytolytic activities by neutrophilia [98]. Studies on intra-tumoral neutrophils have shown conflicting results concerning their correlation with tumor progression. 

Most studies show an unfavorable outcome of high intra-tumoral neutrophil infiltration (e.g., gastric cancer, hepatocellular carcinoma, glioblastoma, bronchioloalveolar carcinoma) [113,114,115,116]. However, higher tumor infiltration of myeloperoxidase (MPO)-positive neutrophils had a favorable prognosis in advanced gastric carcinoma [117], esophageal squamous cell carcinoma [118] and colorectal cancer [119]. 

It seems that a favorable prognosis of tumor-infiltrating neutrophils is related to a concomitant presence of cytotoxic T cells in the tumor tissue [120]. Tumor-infiltrating neutrophils can support adaptive immune responses by recruiting T cells to tumor sites via the secretion of chemokines, such as CCL5, CCL20, CXCL9, CXCL10 and CXCL11 [75,120,121]. Neutrophils can function as antigen-presenting cells and cross-present antigens to activate cytotoxic T cells [122,123,124]. Moreover, neutrophils might lead to activation and recruitment of dendritic cells through secretion of alarmins, such as defensins, cathelicidin, lactoferrin and HMGB1 [125].

### 2.3. Chronic Neutrophilic Inflammation Promotes RAGE-Dependent Carcinogenesis

In human, chronic neutrophilic inflammation has been shown to be involved in the initiation phase of many types of epithelial cancers as well as to contribute to the later phases of cancer development [47,126,127] (Figure 1). The prolonged exposure of epithelial cells to reactive oxygen and nitrogen species (ROS/RNS) produced by inflammatory neutrophils may lead to mutagenesis and the initiation of carcinogenesis [47]. Elevated levels of ROS cause the oxidation of proteins, lipids and nucleic acids [128]. 

Usually, the oxidized proteins become degraded by the 20S proteasome; however, under conditions where the advanced glycation end products (AGEs) are elevated, the bulky structure of AGEs blocks their entry into the proteasomal core [128]. This leads to elevated levels of oxidized and damaged proteins, which promotes further protein modifications [128]. The increased oxidation of lipids and glucose bolsters accelerated formation of AGEs that usually accumulate during aging and in diabetes [128]. AGEs, in turn, activate RAGE and other AGE-receptor complexes [128] resulting in cancer initiation and progression [129,130]. 

Inflammation triggers the upregulation of RAGE, which is a surface molecule important for sustaining inflammation and establishment of chronic inflammatory disorders [131,132,133,134]. RAGE KO mice are protected from the lethal effects of septic shock, which is related to the reduced local inflammation and decreased NFκB activation in the target organs [135]. Blockage of RAGE quenched delayed-type hypersensitivity and inflammatory colitis in murine models by arresting the activation of central signaling pathways and the expression of inflammatory gene mediators [136,137]. RAGE KO mice showed reduced recruitment of neutrophils to inflamed peritoneum, a process that is partly mediated by the interaction of endothelial expressed RAGE with the β-integrin Mac-1 on neutrophils [137]. 

RAGE is upregulated in endothelial cells upon inflammation and the RAGE–Mac-1 interaction acts in concert with ICAM-1–Mac-1 interaction in the recruitment of neutrophils [137]. S100A12 (EN-RAGE) increased VCAM-1 and ICAM-1 expression on endothelial cells in a RAGE-dependent manner [137]. Upregulation of RAGE on epithelial cells may provide outside-in signals that stimulate their proliferation and ultimately cause cancer development [138,139,140]. Thus, RAGE expressed on neutrophils, endothelial cells and epithelial cells is involved in neutrophilic inflammation-induced carcinogenesis. Further description on the role of RAGE in cancer will be discussed in Section 6.

Neutrophils have been documented to promote chronic colitis-associated carcinogenesis in mice [55], which is, in part, mediated by ROS-induced epithelial mutagenesis [141]. In a sustained inflammation model in mice caused by tobacco smoke exposure or nasal instillation of lipopolysaccharide, neutrophil extracellular traps (NETs) were shown to awaken dormant cancer [142]. The NET-associated proteases, neutrophil elastase (NE) and matrix metalloproteinase 9 (MMP-9), cleaved laminin, resulting in the proliferation of dormant cancer cells by activating integrin signaling [142]. In this respect, it is notably that components of NETs (e.g., HMGB1) can interact with RAGE, which elicits proliferative signals in target cells [143].

## 3. Neutrophil Heterogeneity and Subpopulations in Cancer

Several terminologies have entered the literature describing the opposite features of neutrophils in cancer, such as “N1” and “N2” polarization [144], granulocyte myeloid-derived suppressor cells (G-MDSCs) [47,145], neutrophil heterogeneity [65], neutrophil diversity [66], neutrophil plasticity [1,46,146], tumor-associated neutrophils (TANs) [46], tumor-entrained neutrophils (TENs) [60], tumor-educated neutrophils (TENs) [147], tumor-elicited neutrophils (TENs) [148], mature HDNs, immature LDNs (iLDNs) and senescent LDNs (sLDNs) [67]. 

The appearance of all these concepts reflects the influence of the tumor microenvironment on neutrophil function, which undergoes constant dynamic changes in response to the diverse stimuli provided by the tumor cells and other cells in the surroundings. Some of these concepts are overlapping with some nuances. The neutrophil plasticity is the underlying reason for the heterogeneity and diversity, and usually the general neutrophil population is a composite of various neutrophil subpopulations. The TANs and the different forms of TENs can exhibit quite different phenotypes depending on the cancer cell type and the influence of other cells and components in the microenvironment.

The multifaceted characteristics of neutrophils are not surprising considering their nature to be already pre-equipped with a whole battery of substances stored in their many granules [2,5,8,149] and the surface expression of a multitude of chemokine and cytokine receptors [9], pattern recognition receptors [150], C-type lectin receptors [150], Fc receptors [150], carcinoembryonic antigen-related cell adhesion molecule (CEACAM)3 [151], sialic acid-binding immunoglobulin-like lectin (Siglec)-14 [152], leukocyte immunoglobulin-like receptors [153] and complement receptors of the β-integrin family [154] that enable a rapid response to almost any encountered stimuli.

### 3.1. HDN versus LDN

The concepts HDN and LDN come from their different cell buoyancy as defined by a density sucrose gradient. The mature regular HDN population has a cell density greater than 1.080 g/mL contributed by their numerous granules, whereas the LDN population has a density lower than 1.077 g/mL [67,70]. Circulating LDNs are either immature, banded neutrophils released from the bone-marrow prior to full maturation or mature activated neutrophils that have undergone degranulation or senescence. In tumor-bearing mice, immature LDNs might differentiate into HDN in the circulation, and HDNs might turn into senescent LDNs upon activation [67]. 

The major anti-tumor activity is exerted by HDNs, while LDNs has been attributed a pro-tumor role with immunosuppressive properties [67]. The immature LDNs express elevated levels of PD-L1 [155] that suppresses the activities of cytotoxic T cells [155] and NK cells [156]. The IL-6-STAT3 axis was involved in the induction of PD-L1-positive neutrophils [157]. The PD-L1-positive neutrophils were associated with the resolution phase of inflammation [155]. The immature LDNs of 4T1 tumor-bearing mice showed a C/EBPε (CCAAT/enhancer binding protein epsilon) transcriptional signature [158]. C/EBPε is a transcription factor that regulates transition from the promyelocytic stage to the myelocytic stage of neutrophil development, being indispensable for secondary and tertiary granule formation [3].

In healthy mice, most of the circulating neutrophils (95%) are segregated in the high-density fraction, while, in 4T1 breast carcinoma-bearing mice, which are characterized by neutrophilia, the ratio of LDN-to-HDN increases upon tumor progression reaching up to 45–55% LDNs [67]. Costanzo-Garvey et al. [15] noticed that neutrophils in the vicinity of bone metastatic prostate cancer exerted cytotoxic activity against the cancer cells; however, upon tumor progression, the neutrophils failed to elicit cytotoxic effector responses. These findings have raised the hypothesis of an “immunosuppressive switch” where the anti-tumor function of neutrophils is predominant during the early stages of tumor development, while the pro-tumor function is prevailing at the later stages [146].

Notably, in the 4T1 tumor model, tumor cell killing by HDNs from mice with early-stage tumors was similar to that of HDNs from late-stage tumors, indicating that the generation of anti-tumor HDNs still persists despite the dramatic increase in LDNs upon tumor progression [67]. LDNs that have been spontaneously generated from HDN ex vivo, suppressed CD8^+^ T cell proliferation [67]. These LDNs might represent senescent neutrophils. 

An increase in circulating LDNs was also observed in lung and breast cancer patients [67,159]. These LDNs showed increased expression of CD66b, which is a marker of neutrophil activation [67]. Elevated LDN (>10%) correlated with poorer prognosis in late-stage lung cancer patients [159]. The advanced lung patients showed an increase in LDNs expressing the CD66b^+^/CD10^low^/CXCR4^+^/PD-L1^inter^ signature [159], suggesting an increase in the senescent neutrophil population that exhibits tumor-promoting activities.

Costanzo-Garvey et al. [15] observed that the same neutrophil population that was cytotoxic towards the tumor cells were also suppressive to T cells. Likewise, Aarts et al. [160] observed that MDSC activity acquired during neutrophil maturation correlated with the induction of the cytotoxic effector functions of the circulating mature neutrophils. Sagiv et al. [67], however, discerned that these two activities are exerted by distinct neutrophil subpopulations. The regular HDNs were shown to be cytotoxic to tumor cells, while the LDNs were immunosuppressive [67]. As both functions depend on ROS, it is likely that there are some overlapping activities between the different subpopulations.

### 3.2. G-MDSCs

The concept of G-MDSCs was introduced when observing the presence of myeloid-derived suppressor cells in cancer patients that tuned down the immune system [85,86]. The G-MDSCs share many traits with immature LDNs that have been released from the bone-marrow before reaching full maturity [161]. However, not all LDNs are G-MDSCs [158], and G-MDSCs found in cancer may display heterogeneous morphology, including blast-like myelocytes, banded neutrophils as well as mature neutrophils [95]. G-MDSCs also share many pro-tumor features of tumor-associated neutrophils (TANs) [40]. Thus, G-MDSC cannot be categorized into one neutrophil subpopulation but rather is a concept describing a specified neutrophil phenotype.

G-MDSCs promote angiogenesis, produce high levels of MMP9 and augment tumor growth [162]. G-MDSCs show increased NADPH oxidase (NOX2) activity resulting in augmented production of reactive oxygen species (ROS), such as superoxide anion (O_2_^−^_·_), hydrogen peroxide (H_2_O_2_) and peroxynitrite (ONOO^−^_·_) [163,164]. In addition, G-MDSCs show high Arginase 1 (ARG1) activity [163,165] that decomposes l-arginine into urea and L-ornithine [166], and an upregulation of inducible nitric oxide synthase (iNOS/NOS2) that catalyzes the reaction leading to the production of the NO_·_ radical [163]. The enhanced production of ROS together with arginine depletion, nitric oxide radicals, MPO, and inhibitory cytokines contribute to the immunosuppressive features of G-MDSCs [47,163]; however, at the same time, these molecules have anti-tumor activities [41,60,166,167,168].

The MDSC phenotype can be induced by G-CSF [77,169,170,171,172], GM-CSF [173], IL-1β [174,175], IL-6 [176], TNFα [177,178], prostaglandin E2 (PGE2) [179], IL-4 [180], S100A9 [181,182], vascular endothelial growth factor (VEGF) [183], N-formylmethionyl-leucyl-phenylalanine (fMLP) [20], lipopolysaccharide (LPS) [20], ROS [184] and the chemokines CXCL17 [185] and CCL3/4/5 [186]. The CCL3/4/5 chemokines also directly promote tumor growth and angiogenesis [187]. TNFα induced the expression of both S100A8/S100A9 and Receptor for advanced glycation end products (RAGE) on MDSCs, which contributes to their accumulation [178]. The induction of S100A9 is regulated by the STAT3 signaling pathway [182]. 

Interaction of S100A9 with RAGE led to p38 MAPK-mediated chemotaxis of MDSCs, while its interaction with Toll-like receptor 4 (TLR4) induced NFκB-mediated activation of MDSCs [181]. Mice lacking S100A9 mounted potent anti-tumor immune responses due to lack of MDSC induction [182]. Targeted ablation of RAGE in Pdx1-Cre:Kras^G12D/+^ mice limited the development of pancreatic intraepithelial neoplasia lesions with consequent reduced accumulation of MDSCs [188].

### 3.3. “N1” versus “N2” Neutrophils

The concept “N1” was introduced to describe the anti-tumor neutrophil population and “N2” for the pro-tumor neutrophil population [144] in analogy to the anti-tumor “M1” and pro-tumor “M2” macrophages [189]. It should be noted that this nomenclature is a concept made according to a specific function, and thus each neutrophil category might be versatile. The “N1” population can be converted to “N2”, e.g., by TGFβ [75] or after prolonged exposure to G-CSF [190]. Vice versa, the “N2” population can be converted to “N1”, e.g., by IFNβ [53,191]. The G-MDSCs might be considered a subgroup of the “N2” neutrophil population [192]. 

Usually, “N1” neutrophils appear with hypersegmented nuclei with HDN characteristics, whereas “N2” neutrophils often show banded or ring-like nuclei with LDN traits [75] or might be senescent mature neutrophils [67] (Table 3). The “N1” phenotype showed a quite different gene signature than the “N2” phenotype [158,193,194,195]. The anti-tumor “N1” phenotype exhibited increased tumor cytotoxicity, elevated expression of CXCL13, CCL3, CCL6, CXCL10, TNFα and ICAM-1 and low ARG1 content, while the pro-tumor “N2” neutrophils expressed high levels of ARG1, MMP9, VEGF and several cytokines, including CCL2, CCL5, CCL17 and CXCL4 [75,193,194].

### 3.4. Neutrophils at Distinct Stages during Their Lifespan in Circulation Respond Differentially to Stimuli

The heterogeneity of the general circulating neutrophil population seems to be a continuum of distinct activation stages of the neutrophils during their short lifespan as a result of exposure to multiple signals from the microenvironment that act in synergism or antagonism. Only a certain fraction of the circulating neutrophils responds to a stimulus at a given time point, suggesting that they are receptive for a stimulus only at a certain time point during their lifespan in the circulation. Similarly, only a fraction (4–10%) of the HDN population interacts with tumor cells and shows anti-tumor activities at a given time point [196,197]. 

Adrover et al. [25] studied this phenomenon by using Arntl^ΔN^ neutrophils that do not become senescent and remain in the phenotype of “constitutively fresh neutrophils” and Cxcr4^ΔN^ neutrophils that are not reabsorbed back to the bone-marrow and, in such, display a predominant phenotype of “constitutively aged neutrophils”. This research group found that diurnal aging dictates how and when neutrophils migrate into tissues. The freshly released neutrophils responded to the chemokine CXCL2 and migrated into inflamed tissue, while the accumulation of aged neutrophils led to vascular injury and enhanced anti-microbial responses [25].

Other data suggest that subpopulations of neutrophils are activated and differentiated to gain specific functions [27]. For instance, there is a subpopulation of tumor-associated neutrophils whose origin is from CD11b^+^CD15^hi^CD10^−^CD16^low^ immature progenitors that can cross-present antigens, and trigger and augment anti-tumor T cell responses [198]. This subpopulation of neutrophils is triggered by a combined effect of IFNγ and GM-CSF within the tumor [198]. 

Puga et al. [199] observed that neutrophils in the spleen support B cell maturation and antibody production, while Casanova-Acebes et al. [15] found that bone marrow-infiltrating aged (CD62L^low^CXCR4^high^) neutrophils inhibit hematopoietic niches and trigger the release of hematopoietic progenitor cells into the blood stream. Thus, different neutrophil populations might exert specific functions.

## 4. The Delicate Balance between Anti- and Pro-Tumor Neutrophils

### 4.1. Regulation of Anti-Tumor Neutrophils

Factors that can promote the anti-tumor neutrophil phenotype include chemokines (e.g., CXCL2, CXCL5, CXCL8, CCL2, CCL3, CCL5, CXCL12 (SDF-1), CXCL16 and IL-8) alone or together with G-CSF/GM-CSF [60,61,62,200,201], TNFα [200], IFNβ [53], IFNγ [202], IFNγ together with TNFα [203], Resolvin D1 [84], hepatocyte growth factor (HGF) [168] and IL-17 [118] (Table 3). TNFα, CXCL8 and IFNγ could mobilize the extrinsic apoptosis-promoting TRAIL from intracellular stores to the neutrophil cell surface, which is involved in the killing of cancer cells [200,202]. TRAIL might also inhibit tumor growth by preventing the angiogenesis required for their proper growth [204,205]. IFNγ and TNFα reduced the expression of Bv8 and MMP9, while resuming the expression of Rab27a and TRAIL [203]. Rab27a is required for the exocytosis of tertiary and specific granules in neutrophils [206].

In addition, IFNγ and TNFα led to an upregulation of NK-activating ligands, such as RAE-1, MULT-1 and H60, that enhanced the cytotoxic activity of NK cells [203]. The binding of tumor-derived HGF to MET^+^ neutrophils induced the release of nitric oxide (NO_·_), which is cytotoxic to cancer cells [168]. IL-17 was found to potentiate the anti-tumor activity of neutrophils by enhancing the production of cytotoxic molecules, including ROS, MPO, TRAIL and IFNγ [118]. IL-17 also stimulates esophageal squamous cell carcinoma to secrete chemokines CXCL2 and CXCL3, which recruit neutrophils [118]. Resolvin D1, which is involved in the resolution of inflammation, was found to stimulate both the anti-tumor activities of neutrophils and the neutrophil-dependent recruitment of anti-tumor macrophages, thus, enhancing the anti-tumor action [84].

STAT3^KO^ neutrophils from naïve and B16 tumor-bearing mice showed enhanced anti-tumor function towards P815 mouse mastocytoma in comparison to wild-type neutrophils [246]. FasL was increased in STAT3^KO^ neutrophils [246] and contributes to neutrophil cytotoxicity [219]. IFNβ repressed STAT3 activity in tumor-infiltrating IFNβ^KO^ neutrophils resulting in anti-angiogenic and anti-tumor effects [53]. Tumor growth was accelerated in IFNβ^KO^ mice [53]. IFNβ has also a direct effect on cancer cells. It antagonizes the pro-tumor effects of oncostatin M on triple-negative breast cancer stem cells [220]. While oncostatin M induces SMAD3-dependent upregulation of Snail and dedifferentiation of cancer stem cells, IFNβ prevents Snail expression and suppresses tumor growth [220].

### 4.2. Regulation of Pro-Tumor Neutrophils

Factors that promote the pro-tumor neutrophil phenotype include TGFβ [75], VEGF [209,210], IL-11 with FOS-induced growth factor (FIGF/VEGFD) [58], IL-17 [71], IL-35 [211], IL-6 together with G-CSF [211,212], the chemokines CXCL12, CXCL14, CXCL5 and CXCL1 [58], oxysterol [213], hyaluronan fragments [214], GM-CSF [215] and macrophage migration inhibitory factor (MIF) [216] (Table 3). Co-stimulation of neutrophils with IL-6 and G-CSF increased phospho-STAT3 levels, resulting in the upregulation of the pro-angiogenic MMP9 and Bv8 and downregulation of TRAIL [211,212]. 

G-CSF may also stimulate VEGF release from neutrophils [247], thereby, promoting angiogenesis. Increased serum levels of G-CSF and IL-6 have been associated with a poor prognosis in different type of cancer [248,249,250,251,252,253,254]. Priming pro-tumor neutrophils with IFNγ and TNFα could convert them to tumor-suppressing cells even in the presence of G-CSF and IL-6 by restoring the PI3K and p38 MAPK signaling pathways [203]. Hyaluronan was found to activate the TLR4/PI3K/Akt pathway, boost the production of inflammatory cytokines and increase the Mcl-1 levels in the neutrophils resulting in long-lived neutrophils that has lost the ability to kill tumor cells [214]. Long-lived pro-tumor neutrophils were also observed when exposed to IL-6 produced by cancer-derived mesenchymal stem cells [255].

Advanced cancer frequently shows upregulated chemokine expression [187,256], which may contribute to altered neutrophil function in cancer [9,217]. The chemokines are not only important for tumor cell and immune cell trafficking but also have tumor-sustaining activities [217,218,256] and may induce angiogenesis and lymphangiogenesis [218]. The neutrophils themselves are a major source for chemokines [121], such that the initial exposure of neutrophils to tumor-derived chemokines leads to the induction of further chemokine secretion by the neutrophils resulting in a feed-forward regulatory loop [9,41]. The complexity of the chemokine network provides an explanation on how neutrophils encountering a tumor may exert both anti-tumor and pro-tumor properties (Figure 2).

Another example of neutrophil–tumor cell crosstalk is the finding that GM-CSF secreted by human breast cancer cells increases CD11b/CD18 (Mac-1) expression on neutrophils, which interacts with ICAM-1 expressed on the tumor cells, resulting in subsequent neutrophil-mediated transendothelial migration of the tumor cells [257,258]. Sprouse et al. [259] observed that G-MDSCs enhanced the metastatic properties of circulating tumor cells (CTCs) in a positive feedback loop. The CTCs induce ROS production in G-MDSCs through a paracrine Nodal signaling pathway. ROS, in turn, upregulates Notch1 receptor expression in CTCs through the ROS-NRF2-ARE axis, which is then activated by its ligand Jagged1 expressed on G-MDSCs [259].

Engblom et al. [260] observed that lung adenocarcinoma cells release sRAGE into the circulation that activates osteocalcin-expressing osteoblasts in the bone marrow to promote the generation of a specific subset of pro-tumorigenic neutrophils. These neutrophils expressed high levels of sialic acid-binding immunoglobulin-like lectin F (SiglecF), which is not expressed on neutrophils of healthy animals [260]. The SiglecF^hi^ neutrophils expressed higher levels of genes involved in angiogenesis (VEGFA, HIF1a and Sema4d), myeloid cell differentiation and recruitment (CSF1/M-CSF, CCL3 and MIF), T cell suppression (PD-L1 and FCGR2b) and ECM remodeling (ADAM17 and various cathepsins) [260]. The SiglecF^hi^ neutrophils showed increased ROS production compared to SiglecF^low^ neutrophils [260]. The SiglecF^hi^ neutrophils accumulate in lung carcinoma where they can survive for several days [261].

### 4.3. The Anti-Tumor Actions of Neutrophils

To exert direct anti-tumor function, the neutrophils need to be attracted to the tumor cells, interact with the tumor cell and become activated to produce cytotoxic molecules that provide the lethal hit, and the tumor cells need to be susceptible to the cytotoxic molecules. The attraction of neutrophils towards the tumor cells is mediated by chemokines produced by the tumor cells or other cells infiltrating the tumor microenvironment. The chemoattracted neutrophils become activated to produce additional chemokines, thereby, attracting additional neutrophils and other immune cells to the microenvironment in a feed-forward mechanism [41]. 

The activated neutrophils kill the tumor cells by a combined effect of reactive oxygen and nitrogen species, such as hydrogen peroxide (H_2_O_2_), superoxide anion (O_2_^−^_·_), hypochlorous acid (HOCl) and nitric oxide radical (NO_·_), together with FasL and TRAIL [53,60,75,118,167,168,202,219,221,222,223,224,225] (Figure 3). Recently, human neutrophil elastase was found to kill tumor cells by proteolytically liberating the CD95 death domain, which interacts with histone H1 isoforms [262]. Neutrophils might also kill tumor cells by antibody-dependent cellular cytotoxicity (ADCC) [263]. The killing of antibody-opsonized cancer cells by neutrophils was shown to occur by trogocytosis, where the neutrophils retrieve membranes from the tumor cells resulting in the disruption of the cancer cell plasma membrane [264].

Since both the anti-tumor neutrophils and the pro-tumor neutrophils produce ROS while only the former has the propensity to kill tumor cells, it seems that ROS alone is not sufficient to induce tumor cell killing but rather requires additional signals, such as the simultaneous expression of TRAIL [53,200,202]. TRAIL expression is upregulated by IFNβ that converts “N2” neutrophils into “N1” neutrophils [53] and down-regulated when neutrophils are converted into “N2” neutrophils by IL-6 in combination with G-CSF [211,212]. Increased production of ROS can regulate TRAIL signaling in cancer cells by ROS-ERK-CHOP-mediated up-regulation of the TRAIL receptors DR4 and DR5 expression [265]. Furthermore, ROS-induced phosphorylation of Bax at threonine-167, could sensitize melanoma cells to TRAIL-mediated apoptosis [266]. Thus, ROS may increase the susceptibility of cancer cells to TRAIL.

While ROS is crucial for the anti-tumor function of neutrophils [60], ROS can also promote tumor growth by activating the NFκB and PI3K/Akt/mTOR survival signal transduction pathways in the tumor cells [267] and by inducing angiogenesis [268]. In addition, ROS production may contribute to the T cell suppressive activities of MDSCs [20,269], repress NK cell activity [87] and promote carcinogenesis by inducing genotoxic mutations [268,270]. 

ROS production is also important for emergency granulopoiesis [271] and neutrophil extracellular trap (NET) formation [223,272,273,274]. During inflammation, there is an increase in ROS levels in the bone marrow. ROS leads to oxidation and deactivation of phosphatase and tensin homolog (PTEN) resulting in the upregulation of phosphatidylinositol-3,4,5-triphosphate (PtdIns(3,4,5)P3) survival signaling pathways in bone marrow myeloid cells [271]. The production of ROS and nitric oxide might be harmful for healthy tissue and contribute to tissue injury and local microvascular leakage [275]. Thus, neutrophil ROS production might exert both anti- and pro-tumor effects (Figure 4).

Cancer cells are usually more susceptible to neutrophil-mediated killing than normal healthy cells, and neutrophils from cancer patients show, in general, a higher anti-tumor activity than neutrophils from healthy individuals [41,60,225]. Overexpressing of an activated form of the rat sarcoma viral oncogene homolog (Ras) and teratocarcinoma oncogene 21 (TC21) in immortalized mammary epithelial cells was sufficient to sensitize the cells to neutrophil killing [225]. 

Pretreatment of 4T1 breast cancer cells with TGFβ increased their susceptibility to neutrophil-mediated cytotoxicity [68], which might be explained by the more apoptosis-prone mesenchymal phenotype of TGFβ-treated tumor cells [276]. Thus, TGFβ might, on the one hand, prevent the anti-tumor function of neutrophils [75], but might, on the other hand, increase the susceptibility of the tumor cells to the cytotoxic hits. The higher susceptibility of the mesenchymal cells to neutrophils in comparison to epithelial cells of the same tumor [68] is intriguing considering their increased metastatic potential [277,278].

### 4.4. The Pro-Tumor Actions of Neutrophils

Tumor-associated neutrophils may directly or indirectly affect tumor growth and invasion by multiple mechanisms. There is also a complex crosstalk between the neutrophils and tumor cells that fuels tumor cell growth, migration and metastasis. Many of the pro-tumor functions of the neutrophils are related to their wound healing activities [279]. Neutrophils can indirectly affect tumor growth by stimulating angiogenesis [280] and altering the phenotype of endothelial cells [81]. 

The neutrophils secrete several inflammatory, immunoregulatory and angiogenic factors, including NE [226], PR3 [227], CathG [228,229], MMPs [114,230,231], VEGF [53,232,233,234], Bv8 (prokineticin 2) [77,172,235], oncostatin M [236], IL-1β [231], TGFβ2 [237], BMP2 [237] and HGF [116], that modulate the tumor microenvironment and affect tumor growth (Figure 3). The protumor function of neutrophils is also associated with the appearance of MDSCs that suppress T cell functions [47]. 

The suppression of essential anti-tumor T cell functions is, among others, mediated by the production of ROS, peroxynitrite, ARG1, proteases, indoleamine-2,3-dioxygenase (IDO) and NETs as well as the surface expression of PD-L1 and FasL on MDSCs [47,59]. In addition, circulating neutrophils can trap circulating tumor cells at metastatic sites, facilitating their metastatic seeding [54,240].

The secretion of ECM-remodeling enzymes, such as MMP9 (gelatinase B or type IV collagenase), MMP8 (Neutrophil collagenase or Collagenase 2), CathG and NE, paves the way for cancer cell migration and angiogenesis. MMP9 has a direct angiogenic activity, which is inhibited by tissue inhibitor of metalloproteinases (TIMP) [230]. TIMP is degraded by NE resulting in increased MMP9 activity [281]. The remodeling of ECM by neutrophil proteases releases ECM-embedded growth factors, such as bFGF and VEGF that stimulates angiogenesis [267,280,282,283,284]. bFGF also promotes leukocyte recruitment to inflammation by enhancing endothelial adhesion molecule expression [285,286]. Neutrophils can directly induce proliferation of cancer cells through the secretion of NE [287], S100A4 [115], S100A8/A9 [288], FGF2 [82], HGF [116,289], BMP2 [237], TGFβ2 [237] and transferrin [245].

### 4.5. Pro-Tumor Role of Neutrophil Extracellular Traps (NETs)

#### 4.5.1. Regulation of NET Formation

Neutrophil extracellular traps (NETs) are formed when activated neutrophils release their intracellular contents, including DNA, histones and granule components, into the surrounding tissue or circulation [290]. NET formation is dependent on autophagy and is mediated by citrullination of histones to allow for the unwinding and subsequent expulsion of DNA [291]. NET production occurs mainly in activated senescent neutrophils [24]. NET formation is activated by platelet activating factor (PAF) [291] and requires ROS production, NE, CathG and MPO [24,272,292]. 

NETosis occurs in response to tumor-derived factors, such as G-CSF and IL-8 [292,293,294,295]. Upon neutrophil activation, NE translocates from azurophilic granules to the nucleus, where it degrades specific histones, promoting chromatin decondensation [272]. MPO synergizes with NE in driving chromatin decondensation [272]. NET formation frequently occurs in cancer where it promotes tumor growth, metastasis and cancer-associated thrombosis [290,292,294,296,297,298,299]. The production of NETs might also lead to the release of factors that can encourage tumor growth and even metastasis [300].

#### 4.5.2. Trapping of Circulating Tumor Cells by NETs

Microvascular NET deposition can trap tumor cells in the circulation and promote their extravasation to metastatic sites [301,302,303]. The NETs can enwrap and coat the tumor cells, thereby, shielding them from T cell and NK cell cytotoxicity [304]. The prevention of NET formation by DNase I-mediated degradation of NET-DNA, using a NE inhibitor or peptidylarginine deiminase 4 (PAD4)-deficient mice that show defective NETosis, significantly reduced liver metastases of breast and lung carcinoma cells [301]. In addition to trapping the tumor cells, NETs might act as a chemotactic factor that attracts cancer cells [305]. In addition, NETs may increase the vascular permeability at the metastatic site, thereby, facilitating the extravasation of circulating tumor cells [306].

#### 4.5.3. The NET-RAGE Vicious Loop

NETs were found to stimulate the proliferation of pancreatic stellate cells in a RAGE-dependent manner [299]. The RAGE-mediated signals were especially important for the earliest stages of pancreatic cancer development [307,308]. High-mobility group box 1 (HMGB1) associated with NETs interacts with RAGE on the tumor cells, resulting in the activation of NFκB signaling pathways [309]. The activation of RAGE induces IL-8 secretion from the tumor cells, further encouraging the attraction of additional neutrophils [309]. 

Vice versa, RAGE KO neutrophils from tumor bearing animals had a diminished propensity to form NETs [291], suggesting a role for neutrophil RAGE in NET production. Thus, RAGE on neutrophils is important for the pro-tumor phenotype, and RAGE on the tumor cells transmits the pro-survival signals delivered by NETs and other RAGE ligands. HMGB1 binding to RAGE caused an upregulation of both RAGE and TLR4 in bone-marrow-derived macrophages [310]. This research group found a feed-forward regulatory mechanism where HMGB1 induces RAGE-mediated activation of MAPK, which, in turn, promotes TLR4 translocation to the cell surface. Then, signaling through TLR4 caused increased transcription and translation of RAGE [310]. 

Tian et al. [311] observed that HMGB1-DNA complexes promoted the association of RAGE and TLR9, resulting in augmented cytokine release. A recent study by Wang et al. [312] showed that tumor-derived HMGB1 acts on TLR2 to induce CD62L^dim^ neutrophils with a strong ability to produce NETs, which is involved in the lung metastasis of triple-negative breast cancer.

## 5. Neutrophil Recognition of Tumor Cells

A common denominator for many of the pro- and anti-tumor functions of neutrophils is a close cell-cell interaction between neutrophils and tumor cells. Only a few studies have aimed to characterize the molecular mechanisms involved in the neutrophil recognition of tumor cells (Figure 5). The best documented interactions include the Mac-1–ICAM-1, L-Selectin‒Sialomucin, PR3–RAGE and CathG–RAGE couples. The consequences of NET adherence to tumor cells have already been described in Section 4.5. Other interactions have also been observed, such as Notch1 on CTCs with its ligand Jagged1 expressed on G-MDSCs [259], that might contribute to the neutrophil–tumor cell synapse required for the intime interaction between the two cell types.

### 5.1. The Neutrophil Mac-1 Interaction with Tumor ICAM-1

There are some studies that have shown an interaction between neutrophil Mac-1 (CD11b/CD18; Complement Receptor 3) and ICAM-1 (CD54) on certain tumor cells, which facilitated the metastatic seeding of the tumor cells in the liver and the lungs [54,239,240]. Neutrophil interaction with ICAM-1 on the human breast cancer cell MDA-MB-468 enhanced the migratory activity of the tumor cells [313]. Our study using neutrophils from tumor-bearing CD11b KO mice and CD18^dim^ mice, excluded a role for Mac-1 in the anti-tumor function of the neutrophils towards several tumor cell lines. 

Both CD11b KO and CD18^dim^ neutrophils killed the tumor cells tested (LLC lung carcinoma, AT3 breast cancer, 4T1 breast cancer and B16-F10 melanoma cells) as efficiently as wild-type neutrophils (unpublished data). The Mac-1-ICAM-1 interaction is well-known to play an important role in neutrophil rolling and transendothelial migration [314,315], such that mice lacking CD11b or CD18 are expected to show defective neutrophil infiltration of the tumor. Indeed, neutralizing antibodies to CD11b reduced myeloid cell filtration into squamous cell carcinoma and enhanced their response to irradiation [316]. Moreover, using the Apc^Min/+^ spontaneous intestinal tumor model, CD11b deficiency suppressed intestinal tumor growth by reducing myeloid cell recruitment [317].

β-integrin-mediated neutrophil adherence to endothelial cells was shown to suppress ROS production through inhibition of Rac2 guanosine 5′-triphosphatase, an essential regulatory component of NADPH oxidase [318,319]. The suppression of ROS production by β-integrin engagement is proposed to be essential for preventing inappropriate tissue damage during transendothelial migration. Since ROS production is crucial for neutrophil tumor cytotoxicity, β-integrin engagement might transiently antagonize the anti-tumor function of neutrophils.

### 5.2. The Neutrophil L-Selectin Interaction with Tumor Sialomucin and Non-Mucin Ligands

Another molecule that has been shown to be involved in neutrophil interactions with tumor cells is L-Selectin. This molecule is abundantly expressed with ~100,000 copies per neutrophil [320], and it interacts with the vascular sialomucin CD34 [321] and sialomucins on carcinoma cells through its C-type lectin at the amino terminus [241,242]. L-Selectin on leukocytes have been shown to promote metastasis by interacting with both mucin and non-mucin ligands on tumor cells [243,244]. Expression of the L-Selectin ligands sialofucosylated glycans on cancer cells has been linked with poor prognosis and higher rate of metastasis [322]. 

We excluded a role for L-selectin in the anti-tumor function of neutrophils by using soluble L-Selectin containing the extracellular part of the receptor that acts as a decoy molecule and neutralizing antibodies to L-selectin [238]. Soluble TLR4 decoy receptors comprising the extracellular part of TLR4 did not interfere with the neutrophil tumor cytotoxicity. Neutrophils isolated from TLR2 KO and MyD88 KO mice showed even slightly higher cytotoxicity than the wild-type neutrophils (unpublished data).

### 5.3. The Neutrophil Cathepsin G Interaction with Tumor RAGE

Further studies showed that soluble RAGE expressing the extracellular part of RAGE and neutralizing antibodies to RAGE interfered with the anti-tumor function of neutrophils toward several tumor cell lines (e.g., AT3 breast cancer, E0771 breast cancer, LLC lung carcinoma and B16-F10 melanoma), suggesting a role for RAGE in the interaction between the tumor cells and neutrophils [238]. Surprisingly, RAGE KO neutrophils killed the tumor cells to a similar extent as wild-type neutrophils, excluding a role for neutrophil-expressed RAGE in this interaction [238]. 

On the other hand, knocking down or knocking out RAGE in mouse breast and lung carcinoma cells rendered them less susceptible to neutrophil-mediated killing, suggesting that tumor RAGE is the molecule recognized by neutrophils [238]. We and others have demonstrated that human breast cancer and other solid tumors also express elevated levels of RAGE ([140,323,324,325,326] and Figure 6A,B). Using neutralizing antibodies to human RAGE, this molecule was found to be a recognition molecule in the anti-tumor activity of human neutrophils toward human breast cancer cells (Figure 6C). 

Taking into account that RAGE is upregulated during early carcinogenesis (See Section 2.3 and Section 6) and contributes to tumor survival and proliferation, the recognition of tumor RAGE by neutrophils and the consequent tumor cell killing might be considered as an essential tumor immune surveillance mechanism.

When searching for the molecule on neutrophils interacting with tumor RAGE, we discovered CathG as the neutrophil counterreceptor [238]. We further observed that CathG KO neutrophils showed defective cytotoxicity toward RAGE-proficient tumor cells [238], emphasizing the important role of CathG in recognizing RAGE expressed on the tumor cells. This observation was quite surprising in light of the fact that CathG is known to be stored in neutrophil granules [2,5,327,328]. However, it appears that CathG is also expressed on the neutrophil surface [238,329,330]. 

Campbell et al. [330] observed that chondroitin sulfate- and heparan sulfate-containing proteoglycans in the neutrophil plasma membrane are the binding sites for both NE and CathG. Intriguingly, the LDN population expresses even higher levels of CathG on their surfaces than the HDN population [238], suggesting that both HDN and LDN can interact with tumor cells using the same recognition mechanism. The involvement of CathG in neutrophil-mediated tumor cytotoxicity was independent of its proteolytic activity [238]. CathG has previously been shown to be involved in the adhesion of the leukocytes to arterial endothelium in a model of atherosclerosis [331]. This function of CathG was also independent of its proteolytic activity [331]. It would be interesting in this setting to study whether the counterreceptor on the endothelial cells is RAGE.

### 5.4. The Neutrophil PR3 Interaction with Tumor RAGE

Proteinase 3 (PR3) is another neutrophil protease that has been shown to bind to RAGE [227]. In addition to being secreted by the neutrophils, PR3 is allocated on the neutrophil surface [332]. The binding of PR3 to RAGE on prostate cancer caused and outside-in signaling activating the ERK1/2 and JNK1 signaling pathways in the cancer cells resulting in increased cell motility [227]. These signals were induced because of the binding of PR3 to RAGE, without any involvement of its enzymatic activity [227]. RAGE on the prostate cancer cells mediates their homing to the bone marrow, which is rich in PR3-expressing cells [227].

## 6. Pro-Tumor Function of RAGE

### 6.1. General Aspects of RAGE

RAGE is an MHC class III encoded protein belonging to the immunoglobulin (Ig) superfamily that was initially recognized as a receptor for advanced glycation end products (AGEs); however, it rapidly became revealed that this receptor has a multitude of ligands, including S100 proteins such as S100B, S100A4, S100A7, S100A8/A9, S100A11, S100A12, HMGB1, amyloid β peptide, prothrombin, chondroitin sulfate E, heparan sulfate, heparin and the complement C1q and C3a components [140,333,334,335,336,337,338]. Additional RAGE ligands include the neutrophil cationic antimicrobial protein CAP37 and the neutrophil proteases: CathG, NE and PR3 [227,238,339]. RAGE is expressed on several cell types, including immune cells, endothelial cells, fibroblasts, lung epithelial cells, neuronal cells and keratinocytes [128].

### 6.2. Involvement of RAGE in Inflammation-Induced Carcinogenesis

RAGE has been repeatedly shown to be essential for inflammation-induced carcinogenesis, and it is frequently upregulated in cancer [127,129,138,139,140,308,323,324,326,333,340,341,342,343,344,345,346,347,348]. Tumorigenesis is retarded in RAGE KO mice [333,342,345,349,350], and RAGE-knocked-down tumor cells showed defective metastatic properties [238,345,351,352].

### 6.3. RAGE-Induced Signal Transduction Pathways

RAGE is a target gene of the NFκB signaling pathway and signaling through RAGE activates NFκB, thereby, fueling up a feed-forward activation loop [140]. In addition to activation of NFκB, interaction of RAGE with its many ligands, stimulates several pro-survival signal transduction pathways, including Ras-ERK1/2, CDC42/Rac, p38 MAPK, AKT/mTOR and JAK1/2-STAT [128,129,133,138,139,140,353]. S100A14 overexpressed in breast cancer cells promotes metastasis by activating the RAGE-NFκB signaling pathway resulting in the upregulation of CCL2 and CXCL5 expression in the tumor cells [354].

### 6.4. RAGE Ligands with Pro-Tumor Actions

In addition to the direct pro-survival signals delivered by RAGE in cancer cells, RAGE propagates and sustains pro-tumor host inflammatory responses [342]. HMGA1 and HMGB1 binding to RAGE promotes migration, invasion and metastasis of cancer cells [127,355,356,357,358]. The interaction of RAGE with its ligand HMGB1 induces epithelial-mesenchymal transition (EMT) of cancer cells [359,360]. The RAGE ligand S100A7 has been shown to induce EMT in cancer cells [361].

#### 6.4.1. HMGB1

HMGB1 (Amphoterin) is a nuclear non-histone protein that is released from the cell in response to damage or stress stimuli. It is a strong pro-inflammatory protein and tumor promoter that acts on RAGE to activate NFκB and MAP kinase signaling pathways [127,356,358]. The activation of the RAGE-NFκB signaling pathway by HMGB1 induces IL-8 production important for neutrophil recruitment [309]. HMGB1 induces cytokine release from neutrophils and increases the interaction of neutrophils with endothelial cells in a Mac-1 and RAGE-dependent manner, which is required for their subsequent transmigration into inflamed tissue [362]. 

HMGB1 primes vascular cells to upregulate TNFα production and the expression of ICAM-1 and VCAM-1 that strengthen the adhesion of inflammatory cells [363]. The interaction of HMGB1 with tumor RAGE was found to be important for both tumor proliferation and metastasis formation of rat C6 glioma cells and mouse Lewis lung carcinoma cells [356]. Moreover, gastric cancer cell-derived exosomes that contain HMGB1, activate the TLR4/NFκB pathway in the neutrophils, resulting in increased autophagy and induction of the pro-tumor activity [364].

#### 6.4.2. Advanced Glycation End Products (AGEs)

Advanced glycation end products (AGEs) have been found to be expressed in several types of cancer [129] and promote growth, invasion and migration of prostate and breast cancer [365]. The AGE-RAGE interaction leads to increased NADPH oxidase activity, resulting in elevated ROS production [365]. ROS activates NFκB, which upregulates the transcription of iNOS, which produces the nitrogen oxide radical (NO_·_). Superoxide and nitric oxide radicals interact to form peroxynitrite (ONOO^−^_·_), which inactivates functional proteins [128]. Thus, activation of RAGE under inflammatory conditions triggers a vicious signal transduction feedback loop.

#### 6.4.3. S100 Proteins

The S100 proteins are other ligands for RAGE that have been associated with cancer progression [366,367]. Especially, S100A4 has been shown to be overexpressed in various cancer, including breast and pancreatic cancer leading to its nickname “Metastatin” [366]. A strong correlation between S100A4 expression levels and the prognosis of patients with esophageal squamous cell carcinoma, non-small cell lung, melanoma, prostate adenocarcinoma, bladder cancer and gastric cancers has been observed [366]. S100A7, which is frequently overexpressed in ERα^−^ breast cancer, stimulates tumor growth by recruiting MMP9-positive pro-tumor macrophages [345]. 

The S100A8/S100A9 heterodimer highly expressed in neutrophils is involved in inflammatory responses [368]. Low concentration of S100A8/S100A9 promotes tumor growth via a RAGE-dependent mechanism that involves the activation of MAPK and NFκB signaling pathways [369,370,371]. Although S100A9 is pro-apoptotic at high concentration, it is required for colitis-associated cancer development [370]. S100A9 KO mice showed fewer incidences of inflammation-induced colon cancer [370]. 

S100A9 is highly expressed during the acute phase of colitis; however, it is down-regulated by colonic chitinase-3-like 1 (CHI3L1), a pseudo-chitinase that is upregulated during the chronic phase of colitis [372]. CHI3L1 interacts with RAGE to promote intestinal epithelial cell proliferation [372]. These authors proposed that the CHI3L1^high^, S100A9^low^ colonic environment is important for the progression of colitis-induced colon cancer [372]. S100A9 might also interact with Toll-like receptor 4 (TLR4) expressed on tumor cells where it promotes tumor growth [373]. In addition, S100A9 might indirectly promote tumor growth by promoting MDSC-mediated immune suppression [181].

## 7. Pro-Tumor Role of Cathepsin G and Neutrophil Elastase

CathG and NE are two of the four major neutrophil serine proteases that display proteolytic enzymatic activity against extracellular matrix components, such as elastin, fibronectin, laminin, type IV collagen and vitronectin [328] and activates metalloproteases [374], thereby paving the way for neutrophil and tumor cell migration. 

CathG and NE are involved in many physiological and pathophysiological processes and possesses both pro-inflammatory and anti-inflammatory properties depending on the pathophysiological conditions [228,229]. CathG-and NE-mediated proteolysis can either strengthen or suppress the inflammatory responses [228,229,375]. CathG and NE can deactivate receptors and cytokines involved in host defense and inflammation, including the LPS co-receptor CD14 [376], various protease-activated receptors (PARs) [377,378], thrombin receptor [379], TNFα [380] and cytokine receptors [381]. 

In addition, CathG can degrade NKp46 expressed on NK cells and, in such, impair the NKp46-mediated responses of NK cells [382]. On the other hand, CathG can amplify inflammatory responses by processing cytokines. CathG is a chemoattractant for monocytes, osteoclasts, neutrophils and T cells, suggesting that it is important for the transition of inflammatory exudate from neutrophils to mononuclear cells [383,384]. The CathG-induced chemotaxis of monocytes was found to be mediated by proteolytic activation of protease-activated receptor-1 (PAR-1) [384]. 

CathG has also been shown to induce chemotactic activity by interacting with formyl peptide receptor that leads to calcium ion influx, MAPK activation and PKCζ translocation to the cell membrane [385]. CathG plays a role in processing and maturation of chemerin, a chemoattractant that attracts antigen-presenting cells, such as macrophages and dendritic cells [386] and for the proteolytic processing of CXCL5 and CCL15 into more potent chemotactic factors [387,388,389]. CathG facilitated neutrophil infiltration into the pancreas during acute pancreatitis [390].

The CathG/NE KO mice were resistant to endotoxic shock responses, despite TNFα was released to the circulation [391]. This research group found an essential role for the two proteases in the vascular leakage and pulmonary tissue destruction acting downstream to TNFα [391]. CathG has angiotensin-converting properties resulting in a local increase in the Angiotensin II levels in inflamed tissues [392] that leads to destruction of the epithelium barrier [393]. The conversion of Angiotensin I to Angiotensin II by the neutrophil membrane-bound CathG could not be inhibited by the protease inhibitor α1-antichymotrypsin [392]. 

Angiotensin can also be produced by the CathG-mediated activation of prorenin [394]. Angiotensin II is a major regulator of blood pressure and cardiovascular homeostasis, but accumulating data suggest that it also affects cell proliferation, angiogenesis, inflammation and cancer metastasis [395]. Interestingly, El Rayes et al. [79] observed that knocking out CathG and NE in neutrophils or depleting the wild-type neutrophils, prevented pulmonary metastatic seeding of LLC Lewis lung carcinoma cells in an LPS-induced inflammatory lung model. 

The involvement of CathG and NE in the pulmonary metastatic seeding was credited the proteolytic destruction of the anti-tumorigenic factor thrombospondin-1 by the neutrophil proteases [79]. Moreover, CathG has been implicated in the IL-1β processing and secretion from neutrophils, especially under conditions where NFκB is inhibited [396]. The secreted IL-1β encouraged the proliferation of lung cancer cells [396].

## 8. Reconciling the Duality of RAGE and Cathepsin G in Cancer Biology

The involvement of the neutrophil CathG–tumor RAGE interaction in achieving the anti-tumor activity (Section 5.3) is quite intriguing, since tumor RAGE is important for tumor progression and metastasis (Section 6), and there is evidence that CathG is also required for metastasis (Section 7). So how can we reconcile that the two molecules required for metastasis are precisely the same molecules involved in the neutrophil–tumor cell interaction leading to the elimination of the tumor cells?

As we have discussed above (Section 6 and Section 7), both RAGE and CathG have been attributed a central role in cancer progression. Ligation of RAGE induces proliferative signals and may assist in metastatic seeding and survival. Knocking out RAGE prevents carcinogenesis and the formation of metastasis. CathG, in virtue of its ability to promote ECM remodeling, facilitates the migration of both tumor cells and immune cells. 

Knocking out CathG and NE in neutrophils prevented pulmonary metastatic seeding of Lewis lung carcinoma cells in an LPS-induced inflammatory lung model [79]. We also observed that both tumor RAGE and neutrophil CathG are required for metastasis. RAGE KO breast cancer cells showed impaired metastatic seeding capacities [238], and, in a metastatic seeding model where GFP-expressing AT3 breast cancer cells were injected intravenously into mice that have been transplanted with bone marrow (BMT) from either wild-type or CathG KO mice [238], the AT3 cells formed metastases in the lung of wild-type BMT mice, while no metastatic seeding of AT3 was observed in the CathG KO BMT mice (Figure 6D). 

Since neutrophils play an important role in the metastatic seeding of cancer cells in the lung [79], it is likely that the RAGE–CathG interaction is involved in capturing tumor cells at the metastatic site similarly to its involvement in creating the immunological synapse necessary for neutrophil cytotoxicity towards the tumor cells. Whether the neutrophils will promote tumor cell extravasation into the metastatic site or will eliminate the tumor cells will depend on the activation status of the interacting neutrophils and the susceptibility of the tumor cell to the cytotoxic hit at the moment encountering the neutrophils (Figure 7).

The ability of neutrophils to produce ROS is transient since prolonged exposure of the neutrophils to their own ROS production leads to NET production [223], such that the “N1” phenotype will turn into a “N2” phenotype. Although catalase abrogates the tumor cell killing of cancer cells indicative for a central role of hydrogen peroxide in this process [60,225], the pro-tumor neutrophils also produce ROS, suggesting that additional signals are required for anti-tumor activity, such as TRAIL and FasL [53,200,202,203,219]. CathG, NE and ROS were found to act in concert in order to achieve the anti-microbial effect of neutrophils [228,391,397], an observation that raises the question of whether a similar co-operation between CathG and ROS takes place in neutrophil tumor cytotoxicity.

Another question is whether the CathG–RAGE interaction contributes to the cytotoxic hit or if it only strengthens the immunological synapse. CathG has been shown to induce apoptosis of epithelial cells [393] and cardiomyocytes [398]. The apoptosis of epithelial cells was caused by CathG-mediated production of angiotensin II [393]. Cardiomyocytes exposed to CathG showed initial activation of ERK, p38 MAPK and AKT, with the subsequent activation of Caspase 3, cleavage of FAK and AKT, cell detachment and apoptosis [398]. 

Another possibility is that the interaction of CathG with RAGE interferes with the binding of other ligands to tumor RAGE, thereby, altering RAGE-mediated survival signals. Stock et al. [339] observed that CathG could compete with amyloid β1-42 for the same binding site on RAGE. Further studies are required to understand how CathG promotes the anti-tumor action of neutrophils toward RAGE-proficient tumor cells.

## 9. Therapeutic Strategies for Targeting Neutrophils to Strengthen the Anti-Tumor Function

To utilize neutrophils in the combat against cancer, therapeutic strategies should be focused on promoting the anti-tumor function of neutrophils on the expense of the pro-tumor activities. This task is not easy, as these two activities undergo dynamic changes during the lifetime of the neutrophils, and some of the anti-tumor activities are also involved in T cell and NK cell suppression resulting in antagonistic effects on tumor growth (Section 2.1.1 and Section 3.1). 

There are accumulating data indicating that the anti-tumor neutrophils are largely found in the mature HDN population that have just been released from the bone-marrow, while, upon neutrophil senescence, the tumor-promoting activities are predominant (Section 3.1). This suggests that it would be preferable to maintain a short lifespan of the neutrophils avoiding its overactivation to form NETs that are involved in both promotion of tumor cell growth and neutrophil-mediated metastasis (Section 4.5). 

Cytokines that activate neutrophils to an anti-tumor function will ultimately also prolong the longevity of the neutrophils and increase the fraction of pro-tumor neutrophils (Section 4.1 and Section 4.2). The classical example is the 4T1 breast cancer cells that secrete high levels of G-CSF/GM-CSF and CXCL2 that induce the production of both anti-tumor and pro-tumor neutrophils [67]. Future aims should focus on finding a middle way.

### 9.1. Shifting the Neutrophil Activities to an Anti-Tumor Phenotype

Some attempts have been made to shift the balance between anti- and pro-tumor neutrophils in favor of the former. Examples include the inhibition of the signal transduction pathways induced by the immunosuppressive TGFβ and the administration of the immunomodulator IFNβ. In the AB12 mesothelioma cell model, blocking the TGFβ signaling pathway using the type I TGFβ receptor kinase inhibitor SM16 led to an influx of hypersegmented cytotoxic neutrophils that expressed higher levels of pro-inflammatory cytokines [75]. Treatment with SM16 also increased the cytotoxic activity of intra-tumoral T cells resulting in a reduced primary tumor mass [75]. 

In another study, B16F10 melanoma and MCA205 fibrosarcoma cells developed faster-growing tumors with better developed blood vessels in IFNβ-deficient mice compared with syngeneic control mice [53]. The tumors growing in IFNβ-deficient mice showed enhanced infiltration of pro-tumor neutrophils expressing the pro-angiogenic factors VEGF and MMP9 and the homing receptor CXCR4 that is usually upregulated in senescent neutrophils [53]. Treatment of these neutrophils in vitro with IFNβ prevented the expression of the “N2” markers VEGF, MMP9 and CXCR4 [53]. 

Low-dose IFNβ treatment of tumor-bearing mice led to neutrophil polarization towards the anti-tumor N1 phenotype showing elevated Fas and TNFα expression and increased anti-tumor cytotoxicity [191]. Similar changes in neutrophil activation could be observed in melanoma patients undergoing type I IFN therapy [191]. These studies understate the important role of IFNβ in regulating neutrophil function and suggests that IFNβ treatment might have beneficial effects during the early stages of cancer development. In addition to modulating neutrophil function, IFNβ affects other immune cells and has direct anti-tumor activities [399]. The problem of IFNβ therapy is the development of resistance and undesired tumor promoting effects can occur [399].

### 9.2. Targeting the IL-6-STAT3 Axis That Promotes the Pro-Tumor Neutrophil Phenotype

Since STAT3 is involved in the polarization of neutrophils to a “N2” phenotype [53,157,211,212,246], small drugs targeting this pathway are expected to have a beneficial anti-tumor effect. One of the mediators that trigger the STAT3 signaling pathway is IL-6, which plays a role in inflammation-associated cancer [400]. Inhibition of the STAT3-ERK1/2 axis using WP1066, prevented the IL-6-induced pro-migratory and pro-angiogenic properties of neutrophils [255]. WP1066 effectively delayed the progression and invasiveness of bladder cancer in a N-butyl-N-(4-hydroxybutyl) nitrosamine-induced mouse tumor model [401]. 

Bladder cancer could be sensitized to anti-PD-L1 immune therapy by either using anti-IL-6 antibodies or inhibiting the STAT3 pathway with WP1066 [401]. Since soluble IL-6 is shed, among others, from neutrophils by ADAM10- and ADAM17-mediated proteolysis [402,403], inhibition of these proteolytic enzymes is expected to have a beneficial outcome in cancer [404]. Colon cancer formation was impaired in mice lacking ADAM17 or IL-6 [405]. Since ADAM10 and ADAM17 mediate the cell surface cleavage of a large repertoire of substrates that can promote tumor growth [406,407], inhibition of their activities is expected to have general tumor inhibitory activities.

### 9.3. Activation of the Anti-Tumor Neutrophil Function

The pioneering studies performed three to four decades ago used chemokine and cytokine-overexpressing tumor cells to induce tumor rejection that was associated with increased neutrophil infiltration [89,408,409,410,411,412]. The rejection of these tumors was caused by both direct anti-tumor activities of neutrophils and indirectly through neutrophil-induced anti-tumor T cell responses [89,409,410,411,412]. 

A similar approach was recently used by Forsthuber et al. [62], where CXCL5-overexpressing melanoma cells were hindered by neutrophils to form metastases. However, the primary growth of these tumor cells was unaffected by neutrophils. The use of chemokines in the treatment of cancer is not feasible due to multiple and often antagonistic effects of these mediators on tumor progression [413] as well as neutrophil functions [9]. Thus, other strategies should be considered.

The activation of the immune system with BCG (Bacille Calmette-Guérin) has been shown to be beneficial for turning on the immune system in bladder cancer when installed intravesically [414]. This treatment led to the appearance of neutrophils that express membrane-bound TRAIL and secrete large quantities of TRAIL important for the anti-tumor action [415]. These neutrophils also secreted chemokines that attract other immune cells that act in concert to fight cancer [415].

### 9.4. Prevention of NETosis

Another strategy that has been tested is the prevention of NETosis using a peptidyl arginine deiminase 4 (PDA4) inhibitor [303] or disrupting NETs by DNase treatment [292,298,299,301]. Both methods have been shown to reduce metastasis formation and tumor growth [292,299,301,303], emphasizing the importance of NETs in these processes. In a mouse model of Kras-driven pancreatic adenocarcinoma, DNase treatment diminished tumor growth [299]. 

DNase I treatment of MMTV-PyMT tumor-bearing mice, led to a significant reduction in the number of neutrophil-platelet complexes in the kidneys and an improvement of the kidney vasculature [416]. This study demonstrates that neutrophils impair vascular function in the kidneys of tumor-bearing mice by forming NETs [416]. Thus, DNase treatment might be beneficial in reducing not only tumor growth and metastasis but also the vascular toxicities of NETs.

### 9.5. Inhibition of Leukotriene Production

Wculek et al. [76] raised the idea of preventing leukotriene production as a strategy to retard tumor progression. They observed that pharmacological inhibition (e.g., Zileuton) of the leukotriene-generating enzyme arachidonate 5-lipoxygenase (Alox5) in a MMTV-PyMT mammary mouse model, abrogated neutrophil pro-metastatic activity and reduced lung metastasis formation. This therapy relies on tumor dependency on leukotrienes. Indeed, leukotrienes have repeatedly been shown to be involved in tumor-associated inflammation [417]. 

Zileuton prevented polyp formation in the APC^Δ468^ mice by reducing the tumor-associated and systemic inflammation [418]. Tang et al. [419] developed a neutrophil-based nanomedicine based on the natural tropism of neutrophils to inflammatory sites, including tumors. Bis-5-hydroxytryptamine (Bis-5HT) was equipped on nanoparticles loaded with Zileuton to obtain MPO and neutrophil targeting nanoparticles. Bis-5HT oligomerizes and crosslinks with surrounding biological substrates catalyzed by the neutrophil MPO in inflamed tissues. This system was used to show the inhibition of neutrophil-mediated lung metastasis via the sustained release of Zileuton [419].

## 10. Conclusions

Neutrophils can be activated by cancer cells and other cells in the microenvironment to exert pro- and anti-tumor activities. Usually, the two neutrophil phenotypes coexist in cancer together with immunosuppressive G-MDSCs at various ratios that often change during tumor progression from a prominent anti-tumor phenotype to a predominant pro-tumor phenotype. Neutrophils that have acquired anti-tumor activities can later become senescent neutrophils with pro-tumor properties, and immature immunosuppressive neutrophils can turn into anti-tumor neutrophils, indicating that neutrophils show high plasticity. The activation of neutrophils to an anti-tumor phenotype will result in increased neutrophil viability, which, in turn, will lead to a simultaneous expansion of the senescent neutrophil population. 

This continuum of neutrophil activities discourages the use of the “N1” and “N2” terminology that defines two quite different subpopulations. It seems that each neutrophil function can be accomplished within a certain time window during the different maturation stages of the neutrophils after exposure to a stimulus, resulting in a timely neutrophil function within a spectrum of potential activities. It is likely that this scenario is required for the proper resolution of an acute inflammatory response; however, in contrast to wound healing, the inflammatory condition persists in cancer, distorting the normal path of neutrophil function, differentiation and maturation.

There is a great deal of crosstalk between the neutrophils, other immune cells, tumor cells and stromal cells that dictates how the dialog between these cells will restrict or fuel the tumor growth, invasion and metastasis. Tumor cells secrete a range of factors that directly or indirectly affect neutrophil function, and, vice versa, the neutrophils interact with and produce factors that affect the viability and survival of the tumor cells. Sometimes these interactions act in a synergistic feed-forward feedback loop but sometimes in an antagonistic manner. Chemokines can activate the anti-tumor neutrophil function but simultaneously also induce their pro-tumor activities. 

In addition, the chemokines can directly act as growth factors for tumor cells. ROS production is a prerequisite for the anti-tumor function; however, both ROS and RNS can also elicit pro-survival signals and cause mutagenesis that promotes the initiation of carcinogenesis. Excessive ROS and RNS production together with elevated MPO activity result in the suppression of anti-tumor T and NK cells and can lead to local tissue injury. ROS also drives neutrophil senescence resulting in pro-tumor activities accompanied with NETosis. 

The rapid dynamic changes occurring during the lifespan of the neutrophils together with the ever-changing polarization create a high diversity of neutrophil subpopulations, the composition of which will determine the outcome on cancer progression. Anti-tumor neutrophils are characterized by high ROS production together with TRAIL, while pro-tumor neutrophils produce ROS together with nitric oxide radicals, ARG1 and IDO.

Thus far, ECM remodeling enzymes, such as MMP9, CathG, PR3 and NE, have been considered as pro-tumor factors in virtue of their ability to modify the ECM structure, facilitate tumor cell metastasis and release growth factors sequestered to components of the ECM that can further fuel the tumor growth. In addition, MMP9 can promote angiogenesis along with VEGF and Bv8. However, a new function has recently been attributed to CathG and PR3 expressed on the neutrophil surface. Namely, these two enzymes can serve as ligands for RAGE on tumor cells, a function not requiring their proteolytic activity. 

Both HDN and LDN express CathG and PR3 on their surfaces, enabling both populations to use these recognition mechanisms. The CathG–RAGE interaction was found to be important for the neutrophil-mediated killing of RAGE-expressing tumor cells, and paradoxically the same molecules are required for the metastatic seeding of tumor cells. The PR3–RAGE interaction was found to be involved in sequestering circulating RAGE-positive tumor cells to facilitate their infiltration into metastatic sites. Neutrophils can also recognize other molecules on the surface of the tumor cells, such as the Mac-1–ICAM-1, L-selectin–sialomucin and Jagged1–Notch1 interactions that have been shown to facilitate metastatic seedings. 

The interaction between neutrophils and tumor cells is required for both sequestering metastatic cancer cells and for exerting a lethal hit. Thus, the outcome of neutrophil–tumor cell interaction depends on the activation status of the attached neutrophils and the sensitivity of the interacting tumor cells to the lethal hit. Thus, the neutrophils stand as policemen at the crossroad to dictate which tumor cells will die and which will be allowed to enter the metastatic niche (Figure 7).

## Figures and Tables

**Figure 1 cells-10-02486-f001:**
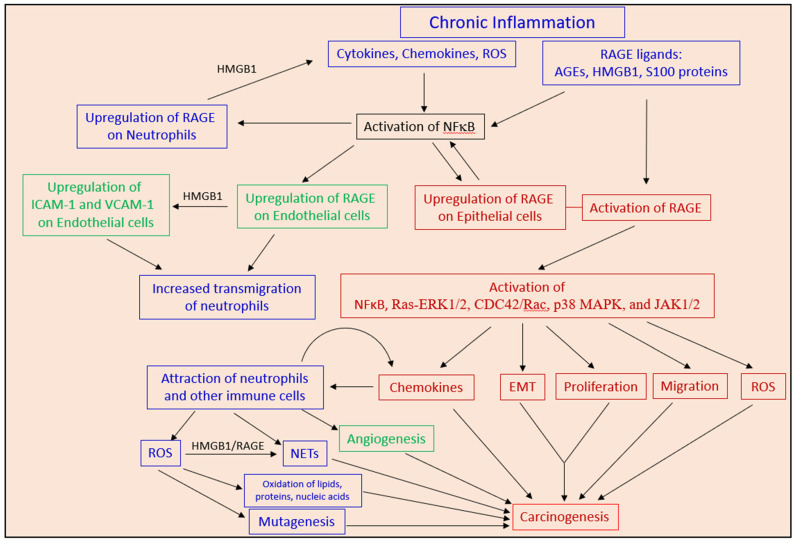
Chronic inflammation-induced carcinogenesis. The figure illustrates pathways in chronic neutrophilic inflammation-induced carcinogenesis that involves the activation of the RAGE signaling pathways in neutrophils (blue boxes), endothelial cells (green boxes) and epithelial cells (brown boxes). The different circuits and pathways are discussed in more details in other parts of the review. Briefly, the chronic inflammation is characterized by neutrophil production of cytokines, chemokines and reactive oxygen species (ROS) that leads to a continuous activation of NFκB in neutrophils, endothelial and epithelial cells, resulting in the upregulation of RAGE, which, in turn, becomes activated by its many ligands present in the inflamed area. Among them, HMGB1, which is released upon tissue injury and NETosis, plays a particular role in activating endothelial cells and in inducing proliferation and migration of epithelial cells. HMGB1 also modulates neutrophil functions. The activated endothelial cells upregulate the adhesion molecules ICAM-1 and VCAM-1 that facilitate neutrophil endothelial transmigration. Activation of RAGE in epithelial cells leads to the production of chemokines that attract more neutrophils and other immune cells, thereby, aggravating the inflammatory process. Other RAGE ligands involved in inflammation-induced carcinogenesis include advanced glycation end products (AGEs) that are proteins or lipids that have become glycated after exposure to excess sugars and S100 proteins, such as S100A4 and S100A7 produced by tumor cells and S100A8/S100A9 produced by neutrophils. The prolonged exposure of the epithelial cells to RAGE ligands, NETs, chemokines, ROS and other stress stimuli, ultimately leads to the initiation of carcinogenesis.

**Figure 2 cells-10-02486-f002:**
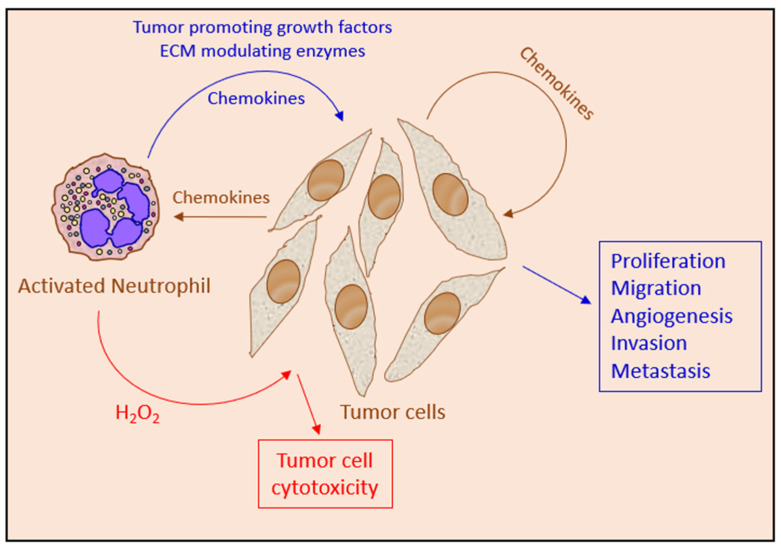
The opposite effects of chemokines in neutrophil tumor biology. Chemokines produced by tumor cells and neutrophils might induce proliferation and migration of the tumor cells. The chemokines also activate neutrophils to produce both tumor cytotoxic factors, such as H_2_O_2_, and pro-tumor factors, such as additional chemokines, growth factors, and ECM modulating enzymes. Thus, the tumor-attracted neutrophil can both exert anti-tumor and pro-tumor actions.

**Figure 3 cells-10-02486-f003:**
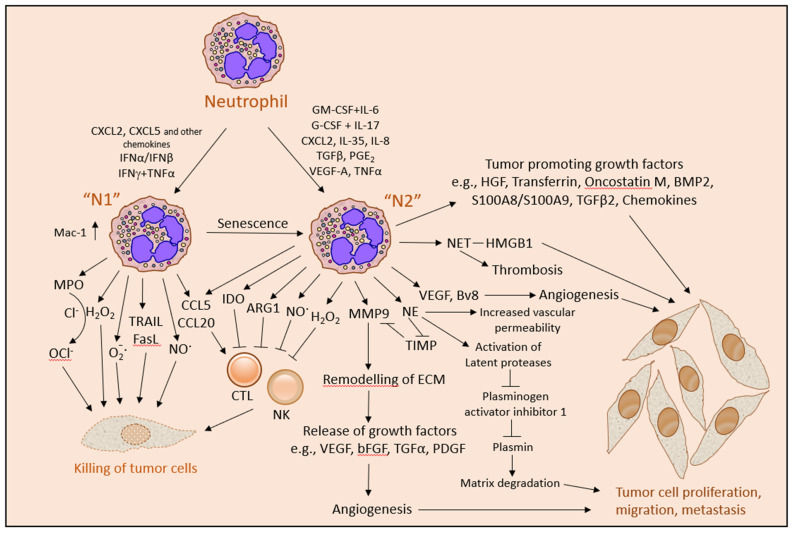
The intricate anti- and pro-tumor functions of neutrophils. The activities of neutrophils can be modulated by a wide range of factors produced by tumor cells, stromal cells, neutrophils and other immune cells. Chemokines and certain cytokine combinations in the presence of interferons can activate the neutrophils to an anti-tumor “N1” phenotype. These neutrophils are important for tumor cell rejection either by a direct tumor cell killing caused by a combination of the cytotoxic molecules H_2_O_2_, O_2_^−^_·_, OCl^−^, NO_·_, TRAIL and FasL, or indirectly by recruiting cytotoxic T lymphocytes (CTL) and NK cells that eliminate the tumor cells. Other cytokine/chemokine combinations can lead to an alternative activation of the neutrophils to acquire a pro-tumor “N2” phenotype that promotes tumor cell proliferation and migration by secreting tumor promoting growth factors and by remodeling the extracellular matrix (ECM). In addition, the “N2” neutrophils promote angiogenesis, which is important for tumor cell expansion, and repress CTL and NK anti-tumor functions. The “N2” population is heterogeneous, composed of immature LDNs, G-MDSCs and senescent neutrophils.

**Figure 4 cells-10-02486-f004:**
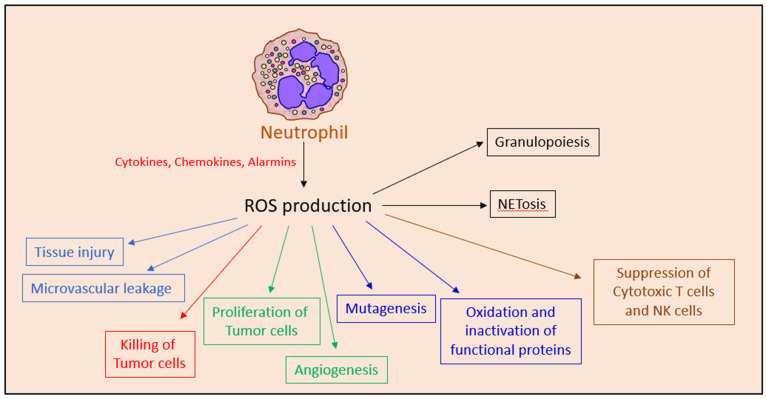
The opposite effects of reactive oxygen species (ROS) in neutrophil tumor biology. ROS produced by neutrophils might either provide the cytotoxic hit resulting in tumor cell death or induce proliferation of the tumor cells. In addition, ROS might lead to tissue injury, angiogenesis and mutagenesis, which further support the cancerous phenotype. On top of this, neutrophil-produced ROS can suppress cytotoxic T cell and NK cell functions necessary for the ultimate tumor cell rejection.

**Figure 5 cells-10-02486-f005:**
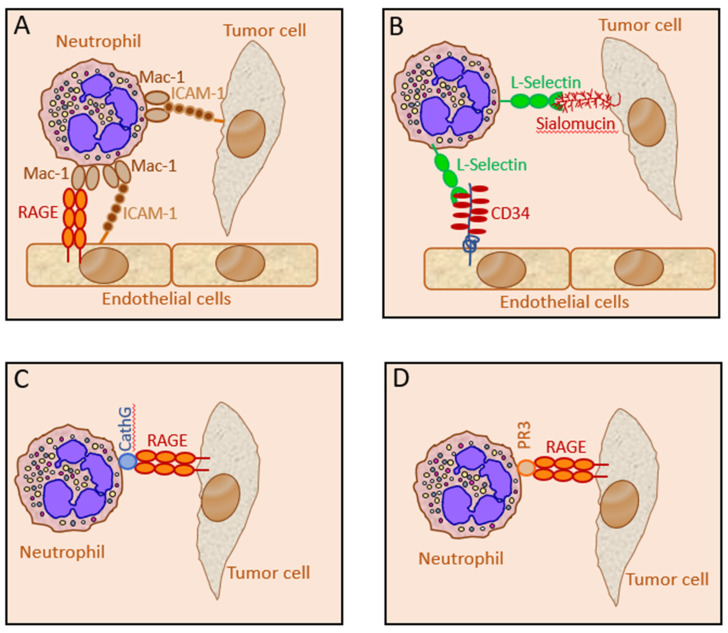
Characterized neutrophil–tumor cell recognition mechanisms. (**A**). Mac-1 (CD11b/CD18) on neutrophils can interact with ICAM-1 on certain tumor cells and endothelial cells. In addition, Mac-1 can interact with RAGE on endothelial cells. The neutrophils form a bridge between the tumor cells and endothelial cells, thus facilitating metastatic seeding. (**B**). L-Selectin on neutrophils can interact with sialomucin on certain tumor cells and CD34 on endothelial cells. Here, the capture of tumor cells by neutrophils that simultaneously interact with endothelial cells enables metastatic seeding. (**C**). CathG on neutrophils can interact with RAGE on tumor cells. This interaction has been shown to be important for executing the killing of RAGE-proficient tumor cells. (**D**). Proteinase 3 (PR3), which is expressed on the neutrophils, can interact with RAGE on tumor cells and induce proliferation of the tumor cells.

**Figure 6 cells-10-02486-f006:**
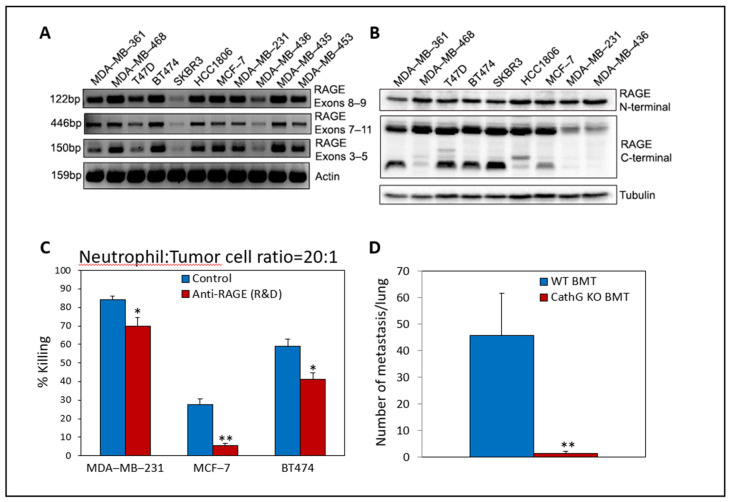
RAGE expression in various human cancer cells and its involvement in neutrophil-mediated cytotoxicity. (**A**). PCR analysis of mRNA levels as determined by using primer pairs for the indicated RAGE exons and primer pairs for β-actin. (**B**). Western blot analysis of protein expression levels as determined by using antibodies to the *N*-terminal part of RAGE (A-9, Santa Cruz, sc-365154), the *C*-terminal part of RAGE (Abcam, ab3611) or α-Tubulin (Sigma, clone DM1A). (**C**). Neutralizing antibodies to human RAGE (R&D) (AF1145; 0.5 μg/mL) inhibited neutrophil-induced tumor cell killing toward MCF-7 breast cancer cells and to a lesser extent toward MDA-MB-231 and BT474 breast cancer cells. *n* = 3 * *p* < 0.05, ** *p* < 0.01. (**D**). AT3 breast cancer cells do not form metastasis in mice, which have been transplanted with Cathepsin G KO bone marrow cells. One hundred thousand GFP-expressing AT3 breast cancer cells were injected intravenously into mice that had been transplanted with either wild-type (WT) or CathG KO bone marrow. The number of GFP-positive metastatic foci in the lungs were counted 8 days later. *n* = 4 for control and *n* = 3 for CathG KO. ** *p* < 0.01. BMT—bone marrow transplanted. These experiments were performed in the laboratory of Prof. Zvi Granot according to the ethics of the Hebrew University’s Institutional Committees.

**Figure 7 cells-10-02486-f007:**
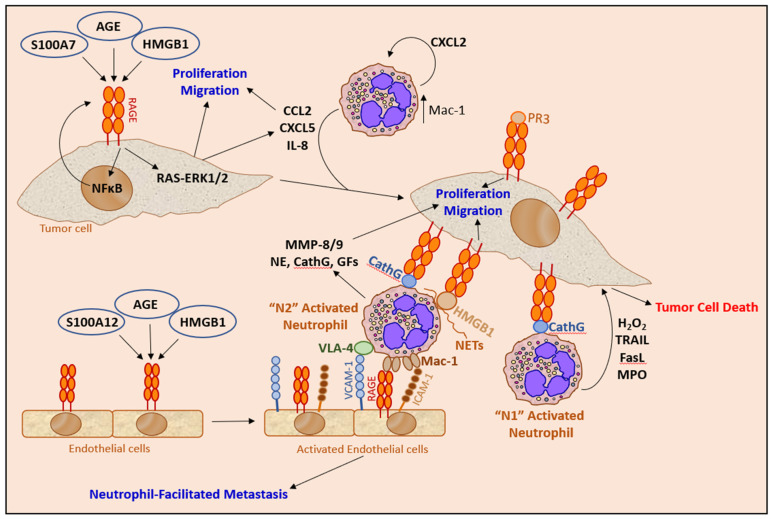
The two faces of the RAGE-Cathepsin G interaction in neutrophil tumor biology. Interaction of RAGE ligands with RAGE on tumor cells leads to proliferation and migration of the tumor cells, as well as the secretion of chemokines that attract and activate neutrophils. The neutrophils can be activated to either anti-tumor “N1” neutrophils or pro-tumor “N2” neutrophils. The interaction of neutrophil-expressed CathG with RAGE on tumor cells facilitates neutrophil-mediated killing of tumor cells. The killing of tumor cells requires simultaneous production of cytotoxic molecules, such as H_2_O_2_, TRAIL, FasL and OCl^−^ produced from H_2_O_2_ by MPO. However, when the tumor cell-interacting neutrophil is alternatively activated to the “N2” phenotype that does not kill the tumor cells, the interaction of neutrophil-expressed CathG with RAGE on tumor cells strengthens the immunological synapse together with other neutrophil–tumor cell interactions (confer Figure 5). Simultaneous interaction of the neutrophils with endothelial cells facilitates the metastatic seeding of the tumor cell into the distant organ. Endothelial RAGE and ICAM-1 interact with Mac-1 (CD11b/CD18) on neutrophils, and endothelial VCAM-1 interacts with α_4_*β*_1_ (VLA-4) on neutrophils. It is still unknown whether CathG on neutrophils can interact with RAGE on endothelial cells. CathG = Cathepsin G, NE = Neutrophil elastase, and GFs = Growth factors.

**Table 1 cells-10-02486-t001:** Examples of tumor models showing pro-tumor neutrophil functions.

Tumor Source	Effect of Neutrophil Elimination	Reference
UV-light induced cancer 4102-PRO that has become resistant to cytotoxic T cells	Elimination of the neutrophils with anti-granulocyte antibodies reduced tumor growth.	[50]
RT7-4bs rat hepatocarcinoma cells	Neutrophils facilitate the attachment of the hepatocarcinoma cells to vascular endothelial cells and increase tumor cell retention in the lungs.	[52]
QR-32 fibrosarcoma	Neutrophil depletion prevented lung metastasis formation without affecting the primary tumor. When conducting the experiment in integrin β2 KO mice that had impaired infiltration of neutrophils into the tumor, a strong reduction in lung metastasis was observed.	[51]
66c14 breast carcinoma cells	Elimination of neutrophils reduced the number of metastases.	[77]
B16F10 melanoma and MCA205 fibrosarcoma cells	Depletion of neutrophils inhibited tumor growth.	[53]
H-59 Lewis lung carcinoma cells	Neutrophil depletion reduced the development of surface liver metastases.	[54]
Chronic colitis-induced colon cancer	Depletion of neutrophils after the last administration of dextran sulfate sodium (DSS), reduced the number and size of the tumors, concomitant with decreased expression of CXCL2, Matrix metalloproteinase 9 and Neutrophil elastase.	[55]
MMTV-PyMT mammary tumor model	Depletion of neutrophils in Rag1-null immune-comprised mice harboring primary tumors during the pre-metastatic stage, led to decreased metastatic seeding.	[76]
A model of invasive intestinal adenocarcinoma (AhCreER; Apc^fl/+^; Pten^fl/fl^ mice).	Depletion of neutrophils suppressed DMBA/TPA-induced skin tumor growth and colitis-associated intestinal tumorigenesis and reduced Apc^Min/+^ adenoma formation.	[78]
A spontaneous breast cancer model (K14^cre^; Cdh1^F/F^; Trp53^F/F^; KEP) mice	IL-17 produced by tumor-infiltrating γδ T cells recruits, expands and activate neutrophils to promote lung metastasis of breast cancer. Neutrophil depletion resulted in significant reduction in both pulmonary and lymph node metastasis without affecting the primary tumor growth.	[71]
LPS-induced lung inflammation model for metastatic seeding of B16-BL6 melanoma and LLC Lewis lung carcinoma cells	Recruitment of neutrophils expressing the inflammatory mediators IL-1β, TNFα, IL-6 and COX2. Depletion of neutrophils suppressed LLC lung metastases. Neutrophil elastase and Cathepsin G degrade Thrombospondin 1, thereby facilitate metastatic seeding in the lung. Mice transplanted with neutrophils deficient for Neutrophil elastase and Cathepsin G showed defective lung metastasis of LLC.	[79]
4T1 subclones selected for high metastasis to the liver, the bone marrow, or the lung	Depletion of neutrophils reduced the liver metastatic burden, but not bone or lung metastatic burdens.	[80]
MMTV-PyVT spontaneous breast cancer model in Col1a1^tm1Jae^ mice resulting in collagen-dense tumors	GM-CSF levels were increased in collagen-dense tumors. Depletion of neutrophils reduced the number of tumors and blocked metastasis in more than 80% of mice with collagen-dense tumors but had no effect on tumor growth or metastasis in wild-type mice.	[57]
Kras^G12D^-driven mouse model of lung cancer	Depletion of Gr1^+^ cells reduced lung tumor growth, reverted immune exclusion and sensitized lesions to anti-PD1 immunotherapy.	[81]
Human HCT-116, LoVo and HT29 colon carcinoma cells	Human colorectal cancer liver metastases and murine gastrointestinal experimental liver metastases are infiltrated by neutrophils. Depletion of neutrophils in established experimental, murine liver metastases led to diminished metastatic growth. Neutrophils contribute to angiogenesis through secretion of FGF2.	[82]
IL-11^+^ and VEGFD^+^ subclones of human MDA-MB-468 breast cancer cells	Depletion of neutrophils prevents lung metastasis without affecting the primary tumor growth. The chemoattractants CXCL12, CXCL14 and CXCL1 that promote the pro-tumor neutrophil phenotype, were found to be secreted by IL-11-responsive mesenchymal stromal cells in the tumor microenvironment.	[58]
Chemically induced cutaneous squamous cell carcinoma (cSCC)	Depletion of neutrophils delayed tumor growth and significantly increased the frequency of proliferating IFNγ-producing CD8^+^ T cells.	[59]
Chronic wound inflammation-induced melanoma in Ras^G12V^ zebrafish larvae	Delaying the development of neutrophils using morpholinos to G-CSF, reduced the number of premetastatic cells.	[56]

**Table 2 cells-10-02486-t002:** Examples of tumor models showing anti-tumor neutrophil functions.

Tumor Source	Effect of Neutrophil Elimination	Reference
Murine ovarian teratocarcinoma	The tumors were rejected in mice treated with *Corynebacterium parvum.* The cytolytic activity was dependent on neutrophils.	[88]
Spontaneous mammary adenocarcinoma TSA	Neutrophils were involved in the rejection of TSA overexpressing various cytokines.	[89]
SBcl2 primary melanoma cells	Depletion of neutrophils enabled the growth and survival of IL-8-overexpressing melanoma cells.	[73]
RM1 mouse prostate cancer cells	Neutrophil depletion prevented rejection of tumor cells induced by adenovirus-mediated IL-12 gene therapy.	[90]
TGFβ blockage of AB12 mesothelioma cells	Depletion of neutrophils in AB12-tumor bearing mice treated with the TGFβR inhibitor SM16, abolished the inhibition of tumor growth caused by SM16.	[75]
4T1 breast cancer cells	Elimination of neutrophils resulted in increased lung metastases without affecting the primary tumor growth.	[60]
RENCA renal carcinoma	Depletion of neutrophils caused an increased rate of metastatic colonization without affecting the primary tumor growth. Human neutrophils displayed a higher cytotoxic activity against poorly metastatic SN12C RCC cells compared to highly metastatic cells. The poorly metastatic SN12C expressed higher levels of CXCL5 and IL-8 that activate the anti-tumor neutrophil function.	[61]
LLC Lewis lung carcinoma	Depletion of neutrophils resulted in enhanced primary tumor growth.	[74]
CXCL5-overexpressing B16F1 melanoma	Overexpression of CXCL5 led to reduced metastasis formation in comparison to control tumor cells. Neutrophil depletion in CXCL5-overexpressing tumor-bearing mice caused increased metastasis formation.	[62]
Prostate cancer cells (C42B, PAIII and LNCaP) injected into the tibia of SCID mice	Prostate cancer cells secrete factors that activate neutrophils to kill the tumor cells. Neutrophil depletion led to increased tumor growth in the bones.	[91]
E0771 breast cancer cells	In NK cell-deficient mice, G-CSF-expanded neutrophils showed an inhibitory effect on the metastatic colonization of breast tumor cells in the lung. In NK cell-competent mice, neutrophils facilitated metastatic colonization in the same tumor models.	[87]

**Table 3 cells-10-02486-t003:** Characteristics of anti- and pro-tumor neutrophils. The table emphasizes some specific traits that have been attributed to the activation and function of anti- versus pro-tumor neutrophils. Concerning the heterogeneous “N2” population, some of the traits are related to the immature LDNs (labeled with *) or the senescent mature neutrophils (labeled with **). Most studies on “N2” neutrophils have not discerned between the different “N2” subpopulations, and thus the general concept is provided. The section number in which the subject is discussed is mentioned in parentheses.

	Anti-Tumor Neutrophils (“N1”)	Pro-Tumor Neutrophils (“N2”)	References
Neutrophil subpopulation (Section 3)	Mainly mature HDN with hypersegmented nuclei	LDN; G-MDSC; immature neutrophils with banded or ring-like nuclei; senescent neutrophils	[46,47,60,67,144,145]
Induction of the phenotype (Section 4.1 and Section 4.2)	CXCL2 + GM-CSF/G-CSF Diverse cytokines (e.g., CCL2, CCL3, CCL5, CXCL5, CXCL12 (SDF-1) and CXCL16) IFNβ IFNγ + TNFα IL-17 Resolvin D1	CXCL5, CXCL8, CXCL17, CCL3/4/5 fMLP, TNFα or LPS G-CSF, GM-CSF IL-1β, IL-4, IL-6, IL-11, IL-35 IL-6 + GM-CSF IL-17 + G-CSF PGE_2_, S100A8/S100A9, VEGF TGFβ Hyaluronan fragments MIF Oxysterol Cancer-specific peptide of Vacuolar-ATPase a2 isoform (a2NTD)	[9,14,20,53,58,60,61,62,68,71,75,77,84,116,118,159,168,169,170,171,172,173,174,175,177,178,179,180,181,182,183,184,185,186,194,200,201,202,203,207,208,209,210,211,212,213,214,215,216,217,218]
Characteristics of the subpopulations (Section 3, Section 4.3 and Section 4.4)	Increased Mac-1 and ICAM-1 expression. CD62L^high^CXCR4^low^ Produce H_2_O_2_, O_2_^−^_·_, OCl^−^, FasL and TRAIL, which are cytotoxic to tumor cells Secrete CXCL13, CCL3, CCL6, CXCL10, TNFα	CD62L^high^CXCR4^low^ * CD62L^low^CXCR4^high^TLR4^high^ ** Increased Mac-1 and ICAM-1 expression ** Increased PD-L1 expression * Produce H_2_O_2_, O_2_^−^_·_, OCl^−^, NO_·_, ONOO^−^_·_, ARG1, IDO Secrete MMP8/9, NE, CathG, VEGF, Bv8, FGF2 HGF, CCL2, CCL5, CCL17, CXCL4 LOX-1, FATP2, COX2, iNOS IL-1β, TNFα, IL-6, Oncostatin M, Transferrin NET production ** Represses the cytotoxic activity of T and NK cells *	[15,24,47,53,59,60,61,75,77,114,116,118,155,167,168,172,202,219,220,221,222,223,224,225,226,227,228,229,230,231,232,233,234,235,236,237]
Recognition of tumor cells (Section 5)	Cathepsin G-RAGE interaction	Mac-1–ICAM-1 interaction L-selectin-sialomucin interaction PR3-RAGE interaction	[54,227,238,239,240,241,242,243,244]
Activated intracellular signaling pathways of the subpopulations (Section 3, Section 4.1 and Section 4.2)	PI3K and p38 MAPK signaling pathways Src kinase Upregulation of Rab27a	STAT3 activation JAK/STAT5β Immature LDNs possess a C/EBPε transcriptional signature	[21,53,157,158,203,245,246]

## Data Availability

Any raw data relevant to this manuscript are available upon reasonable request.

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
