# Peer review of "Leveling Up the Controversial Role of Neutrophils in Cancer: When the Complexity Becomes Entangled"

_cells, 2021, doi:10.3390/cells10092486_

Round 1

Reviewer 1 Report

The manuscript by Sionov is written very well. It covers the current studies in the field of neutrophil biology in cancer, but it lacks deep discussions, intellectual/professional inputs, and re-analysis. I recommend revising the manuscript to overcome the shortages and to provide examples that future researchers must investigate to help with the development of new therapeutic options for cancer treatment. 

A couple of examples:

- Line 51-52: “Also, the neutrophils in the metastatic site might have different characteristics than those in the circulation and those at the primary tumor site [28-31].” Is there anything in the literature that suggests why this can be the case? Some examples of specific differential activation pathways would be helpful to include here.

- Line 56-58. “the simultaneous presence of myeloid-derived suppressor cells (MDSC) that tune down the activities of both cytotoxic T and NK cells, can overweigh the anti-tumor neutrophil function [34,35].” The sentence needs to clarify if the presence of MDSCs influences the behavioural and phenotypic patterns of neutrophils to induce T and NK cell suppression.

- Section 2.1. Considering this is a subtopic under "Neutrophil heterogeneity in cancer", there is a lack of details to support how HDN and LDN differ in various cancer settings.

- Meylan group (Nature Communications, 2020) criticized the methods employed to deplete neutrophils in vivo. This should be mentioned in the manuscript since the author entirely relied on studies that eliminate neutrophils to prove their findings.

- Evidence showed that neutrophil polarization is possible, at least in vitro. What are the author’s thoughts regarding these observations? Is there any potential to take advantage of neutrophil polarization to develop therapeutic agents for targeting cancer and other neutrophil mediated diseases?

Author Response

The manuscript by Sionov is written very well. It covers the current studies in the field of neutrophil biology in cancer, but it lacks deep discussions, intellectual/professional inputs, and re-analysis. I recommend revising the manuscript to overcome the shortages and to provide examples that future researchers must investigate to help with the development of new therapeutic options for cancer treatment. 

A couple of examples:

- Line 51-52: “Also, the neutrophils in the metastatic site might have different characteristics than those in the circulation and those at the primary tumor site [28-31].” Is there anything in the literature that suggests why this can be the case? Some examples of specific differential activation pathways would be helpful to include here.

The following text has been added to clarify this issue: "For instance, neutrophils isolated from the primary tumor of 4T1 breast carcinoma barely exhibited anti-tumor activities, while those isolated from the lungs of the same animals showed anti-tumor activities to a similar extent as the circulating neutrophils [68]. It has been suggested that high levels of TGFβ in the primary tumor prevent the anti-tumor function of neutrophils and promote the appearance of an immunosuppressive neutrophil population [68,75]. The TGFβ level is anticipated to be much lower in the pre-metastatic lung which thus enables the actions of anti-tumor neutrophils [68]. However, in the MMTV-polyoma middle T antigen (PyMT) mammary tumor mouse model, neutrophil recruitment to the pre-metastatic lung could specifically support metastatic initiation through neutrophil-derived leukotrienes that promotes the growth of a subpopulation of cancer cells [76]. Again, we see that the cancer regulating activities of neutrophils are complex, full of dualities which will be further discussed in this review."

- Line 56-58. “the simultaneous presence of myeloid-derived suppressor cells (MDSC) that tune down the activities of both cytotoxic T and NK cells, can overweigh the anti-tumor neutrophil function [34,35].” The sentence needs to clarify if the presence of MDSCs influences the behavioural and phenotypic patterns of neutrophils to induce T and NK cell suppression.

Text has been added and modified to clarify this issue: "As will be discussed in Section 3, the neutrophils in cancer constitute a heterogeneous population of both anti- and pro-tumor neutrophils as well as granulocyte myeloid-derived suppressor cells (G-MDSCs) [7,48,65-67]. The ratio and locations of the different neutrophil subpopulation might dictate the net effect of neutrophils on tumor progression and metastasis."  Tumor rejection is achieved by a combined effect of direct anti-tumor activity of neutrophils, neutrophil-induced anti-tumor T cell responses, and anti-tumor NK cell activities [83]. Neutrophils might also modulate the anti-tumor function of macrophages [84]. However, the simultaneous presence of G-MDSCs that tune down the activities of both cytotoxic T and NK cells, might overshadow the anti-tumor neutrophil function in various experimental settings [85,86]."

- Section 2.1. Considering this is a subtopic under "Neutrophil heterogeneity in cancer", there is a lack of details to support how HDN and LDN differ in various cancer settings.

The following text has been added to illustrate this issue: " In healthy mice, most of the circulating neutrophils (95%) are segregated in the high-density fraction, while in 4T1 breast carcinoma-bearing mice which are characterized by neutrophilia, the ratio of LDN-to-HDN increases upon tumor progression reaching up to 45-55% LDNs [67]. Costanzo-Garvey et al. [15] noticed that neutrophils in the vicinity of bone metastatic prostate cancer exerted cytotoxic activity against the cancer cells, but upon tumor progression, the neutrophils failed to elicit cytotoxic effector responses. These findings have raised the hypothesis of an "immunosuppressive switch" where the anti-tumor function of neutrophils is predominant during the early stages of tumor development, while the pro-tumor function is prevailing at the later stages [146].

Notably, in the 4T1 tumor model, tumor cell killing by HDNs from mice with early-stage tumors was similar to that of HDNs from late-stage tumors, indicating that the generation of anti-tumor HDNs still persists despite the dramatic increase in LDNs upon tumor progression [67]. LDNs that have been spontaneously generated from HDN ex vivo, suppressed CD8+ T cell proliferation [67]. These LDNs might represent senescent neutrophils. An increase in circulating LDNs was also observed in lung and breast cancer patients [67,159]. These LDNs showed increased expression of CD66b which is a marker of neutrophil activation [67]. Elevated LDN (>10%) correlated with poorer prognosis in late-stage lung cancer patients [159]. The advanced lung patients showed an increase in LDNs expressing the CD66b+/CD10low/CXCR4+/PD-L1inter signature [159], suggesting an increase in the senescent neutrophil population that exhibits tumor-promoting activities."

- Meylan group (Nature Communications, 2020) criticized the methods employed to deplete neutrophils in vivo. This should be mentioned in the manuscript since the author entirely relied on studies that eliminate neutrophils to prove their findings.

The following text has been added to address this query: " The conclusions of these studies are based on the use of antibodies that eliminate neutrophils, but it should be kept in mind that the neutrophils are continuously replenished from the bone-marrow, and the newly released neutrophils which have resisted anti-Ly6-mediated depletion, may still be functional [63]. Also, there is no possibility to completely remove all neutrophils because this can lead to life-threatening infections. Nevertheless, reducing the neutrophil number in tumor-bearing mice was sufficient to alter the ability of the cancer cells to metastasize, whether it is an increase or a decrease in the metastatic capability. "

- Evidence showed that neutrophil polarization is possible, at least in vitro. What are the author’s thoughts regarding these observations? Is there any potential to take advantage of neutrophil polarization to develop therapeutic agents for targeting cancer and other neutrophil mediated diseases?

This is a good question. A new section (Section 9) that discusses potential approaches to modulate neutrophil polarization has been added to the manuscript.

Reviewer 2 Report

The author described in detail about the conflicting findings regarding the neutrophils-mediated actions on carcinogenesis. This review article is well written in each paragraph; however, it would be better to consider paragraph composition to make it easier to understand the author’s claim.

  • Introduction section actually consists of two compartments. In the former of introduction section, the author had better show the background of the topics and what points the author would like to focus on briefly. The last half of introduction section would be better to be divided the following paragraph, such as “Overview of tumor models showing pro-/anti-tumor neutrophil functions”.
  • Like the Introduction section, the composition of “2. Neutrophil Heterogenicity in Cancer” should be considered. The first part would be better to be divided such as “Neutrophil kinetics in tumor initiation, development, and metastasis”
  • That is also in “5. Pro-Tumor Function of RAGE”.
  • Figure 6 should be sophisticated.
  • The last half of Conclusion section including Figure 7 should be combined with the paragraph “7. Reconciling The Duality of RAGE and Cathepsin G in Cancer Biology”.

Author Response

The author described in detail about the conflicting findings regarding the neutrophils-mediated actions on carcinogenesis. This review article is well written in each paragraph; however, it would be better to consider paragraph composition to make it easier to understand the author’s claim.

Several paragraphs have now been divided into sub-paragraphs.

  • Introduction section actually consists of two compartments. In the former of introduction section, the author had better show the background of the topics and what points the author would like to focus on briefly. The last half of introduction section would be better to be divided the following paragraph, such as “Overview of tumor models showing pro-/anti-tumor neutrophil functions”.

The Introduction section has accordingly been rearranged to include background information of neutrophils followed by a section (Section 2) describing evidence for a role of neutrophils in cancer.

Like the Introduction section, the composition of “2. Neutrophil Heterogenicity in Cancer” should be considered. The first part would be better to be divided such as “Neutrophil kinetics in tumor initiation, development, and metastasis”

Text has been added in this section (currently Section 3) to better clarify the issues. Since this section deals with various subpopulations of neutrophils in cancer,  I have changed the title to: "Neutrophil Heterogeneity and Subpopulations in Cancer".

  • That is also in “5. Pro-Tumor Function of RAGE”.

This section is now divided into sub-paragraphs.

  • Figure 6 should be sophisticated.

The figure (Currently Figure 1) illustrates RAGE-mediated signal transduction pathways in chronic neutrophilic inflammation-induced carcinogenesis. Other issues are dealt with elsewhere. Some additional complexity has been added to the figure.

  • The last half of Conclusion section including Figure 7 should be combined with the paragraph “7. Reconciling The Duality of RAGE and Cathepsin G in Cancer Biology”.

The conclusion section has now been rewritten.

Reviewer 3 Report

This review article describes a complex mode of mutual interaction between neutrophils, other types of immune cells, stromal cells and tumor cells, which determines the pro- or anti-tumor activities.  The author apparently succeeds in covering such an interesting but complex area by citing references in an unbiased fashion.  Many readers may be interested in this article and therefore this may be worthy of publication. 

The author explained how the same ROS exerts the opposite effects by focusing on N1, N2 and TRAIL.  If the author would strengthen N1 and N2 dichotomy whenever possible, for instance when the role of L-selectin in metastasis is discussed or when HDN and LDN are discussed, throughout the article, then readers may find some clues from the article more easily. 

There are concerns as follows, which should be answered appropriately by the author.

(1) In the end of the first paragraph of the introduction in line 44 to 47, the author stated that there is no single clue that can explain these contradictory effects.  The readers may get lost here, because they may not understand the purpose of the article.  The author may be asked to modify the sentence in a positive way.

(2) In line 71, “it” may be changed to “its”.

(3) In line 76 to 77, “when taken into account that …” may be changed to “when we take into account that …”.

(4) Table 3 should include CD62L, CXCR4, cathepsin G and others together with section numbers where these molecules are discussed, and then link these molecules with N1 and N2. 

(5) Since N2 consists of immature neutrophils and senescent neutrophils, the author may be advised to differentiate these two neutrophils in the case of NET formation which occurs in senescent neutrophils among N2.

(6) In line 266, “Longevity may be changed to “Long-lived”.

(7) In line 431 to 446, RAGE on neutrophils and that on tumor cells are not discussed separately, and so it is rather hard to follow.  The similar problems are often seen in the article.   

(8) The paragraphs under subheading 5.1 may be changed to a different location.

(9) In line 721, “were” may be changed to “where”.

(10) In line 724 to 725, the first sentence under subheading 7 is rather hard to follow, because previous sections focus on pro-tumor activity of an interaction of Cathepsin G with RAGE.

(11) In line 742, “CathG BMT mice” may be changed to “CathG KO BMT mice”.

Author Response

This review article describes a complex mode of mutual interaction between neutrophils, other types of immune cells, stromal cells and tumor cells, which determines the pro- or anti-tumor activities.  The author apparently succeeds in covering such an interesting but complex area by citing references in an unbiased fashion.  Many readers may be interested in this article and therefore this may be worthy of publication. 

The author explained how the same ROS exerts the opposite effects by focusing on N1, N2 and TRAIL.  If the author would strengthen N1 and N2 dichotomy whenever possible, for instance when the role of L-selectin in metastasis is discussed or when HDN and LDN are discussed, throughout the article, then readers may find some clues from the article more easily. 

The "N1" and "N2" dichotomy is general concepts for anti- and pro-tumor neutrophils, respectively, both being composed of heterogeneous neutrophil populations that undergo constant dynamic changes. Therefore, I found it better to use the concepts "anti-tumor" and "pro-tumor" neutrophils throughout the text. In some places where the reference specifies the "N1" or "N2" phenotype, this terminology has been adapted in the text.

There are concerns as follows, which should be answered appropriately by the author.

  • In the end of the first paragraph of the introduction in line 44 to 47, the author stated that there is no single clue that can explain these contradictory effects.  The readers may get lost here, because they may not understand the purpose of the article.  The author may be asked to modify the sentence in a positive way.

The first part of the sentence has been changed to "There is no simple explanation for these contradictory effects"

(2) In line 71, “it” may be changed to “its”.

Corrected.

(3) In line 76 to 77, “when taken into account that …” may be changed to “when we take into account that …”.

Done.

(4) Table 3 should include CD62L, CXCR4, cathepsin G and others together with section numbers where these molecules are discussed, and then link these molecules with N1 and N2. 

These molecules have now been included in the table, and references have been added and well as reference to the Sections describing these effects.

(5) Since N2 consists of immature neutrophils and senescent neutrophils, the author may be advised to differentiate these two neutrophils in the case of NET formation which occurs in senescent neutrophils among N2.

Asterisks have been added to the Table when specific traits have been attributed to either immature LDNs or senescent neutrophils. Many studies have used the N2 terminology in general without specifying which N2 subpopulation they are describing.

(6) In line 266, “Longevity may be changed to “Long-lived”.

Corrected.

(7) In line 431 to 446, RAGE on neutrophils and that on tumor cells are not discussed separately, and so it is rather hard to follow.  The similar problems are often seen in the article. 

Mutual interplays between neutrophils and cancer cells are discussed at various places in the text, and therefore these are discussed together. Some changes have been done in the text to avoid potential confusions.

(8) The paragraphs under subheading 5.1 may be changed to a different location.

The paragraph has now been moved to Section 2.3.

(9) In line 721, “were” may be changed to “where”.

Corrected.

(10) In line 724 to 725, the first sentence under subheading 7 is rather hard to follow, because previous sections focus on pro-tumor activity of an interaction of Cathepsin G with RAGE.

Now I have added the Section numbers where the different issues are discussed. This should put the sentences in context.

(11) In line 742, “CathG BMT mice” may be changed to “CathG KO BMT mice”.

Done.

Round 2

Reviewer 1 Report

The author revised the manuscript based on my comments. Generally, my points have been addressed.